# PhyScensis: Physics-Augmented LLM Agents for Complex Physical Scene Arrangement

**Yian Wang**[1,2*]   **Han Yang**[1*]   **Minghao Guo**[3]   **Xiaowen Qiu**[2]
**Tsun-Hsuan Wang**[2]   **Wojciech Matusik**[3]   **Joshua B. Tenenbaum**[3]   **Chuang Gan**[1,4†]

[1]UMass Amherst   [2]Genesis AI   [3]MIT   [4]MIT-IBM Watson AI Lab
{yianwang,hanyang,chuanggan}@umass.edu
guomh2014@gmail.com
{xiaowen.qiu,johnson}@genesis-ai.company
{wojciech,jbt}@mit.edu

## Abstract

Automatically generating interactive 3D environments is crucial for scaling up robotic data collection in simulation. While prior work has primarily focused on 3D asset placement, it often overlooks the physical relationships between objects (e.g., contact, support, balance, and containment), which are essential for creating complex and realistic manipulation scenarios such as tabletop arrangements, shelf organization, or box packing. Compared to classical 3D layout generation, producing complex physical scenes introduces additional challenges: (a) higher object density and complexity (e.g., a small shelf may hold dozens of books), (b) richer supporting relationships and compact spatial layouts, and (c) the need to accurately model both spatial placement and physical properties. To address these challenges, we propose PhyScensis, an LLM agent-based framework powered by a physics engine, to produce physically plausible scene configurations with high complexity. Specifically, our framework consists of three main components: an LLM agent iteratively proposes assets with spatial and physical predicates; a solver, equipped with a physics engine, realizes these predicates into a 3D scene; and feedback from the solver informs the agent to refine and enrich the configuration. Moreover, our framework preserves strong controllability over fine-grained textual descriptions and numerical parameters (e.g., relative positions, scene stability), enabled through probabilistic programming for stability and a complementary heuristic that jointly regulates stability and spatial relations. Experimental results show that our method outperforms prior approaches in scene complexity, visual quality, and physical accuracy, offering a unified pipeline for generating complex physical scene layouts for robotic manipulation. More qualitative results are on physcensis.github.io.

## 1 Introduction

The creation of large-scale training data has become a key driver in advancing robotics and embodied AI. A prominent line of recent work highlights simulation-based data generation as an effective strategy (Deitke et al., 2022; Wang et al., 2023b;a; Ha et al., 2023; Dalal et al., 2023; Wang et al., 2024b; Pfaff et al., 2025; Wang et al., 2025), as simulation offers a scalable and cost-efficient means of data collection. Among the various forms of data required for training embodied agents, constructing diverse and complex environments for robot manipulation tasks is particularly crucial (Wang et al., 2023b; Yang et al., 2024c; Wang et al., 2024b; Pfaff et al., 2025).

Prior efforts have approached this problem using procedural, rule-based generation (Deitke et al., 2022; Raistrick et al., 2024), where human experts manually design rules for scene construction. While effective in controlled cases, these methods are inherently constrained to the scenarios envisioned by the designers. Another line of work trains models on large-scale 3D scene

---

*Equal contribution
†Corresponding author

**(a) Complexity**

(1)  (2)  (3)  (4)

**(b) Controllability**

3 stacks of books, each with 3-4 books where the books in each stack are visibly different from each other but all have roughly the same color within each stack.

An organized dining table set up for 4 people.

more compact

messier

(1)  (2)

**(c) Diversity**

A deck offloading scene with a hand truck and stack of boxes.

A sink with 2 stacks of plates and other tableware.

A working desk of a computer science PhD student.

A box full of books.

Figure 1: We present *PhyScensis*, an agentic framework that incorporates a physics engine for physical scene arrangement. PhyScensis is: (a) capable of generating complex scenes with high object density and intricate physical interactions; (b) highly controllable, with with strong text-following abilities; and (c) adaptable to diverse, open-vocabulary scenarios.

datasets (Paschalidou et al., 2021; Feng et al., 2024; Tang et al., 2023; Yang et al., 2024b; Pfaff et al., 2025). Although such methods enable learning-based generalization, they remain limited by the availability of datasets (e.g., Fu et al. (2020)), which provide only sparse coverage of detailed small-object placement. Moreover, both rule-based and data-driven approaches typically operate on fixed asset libraries, hindering their ability to support open-vocabulary generation.

Recent work has also explored LLM-based agent frameworks for open-vocabulary scene generation (Wen et al., 2023; Yang et al., 2024c; Wang et al., 2024b; Ling et al., 2025; Wang et al., 2023b; Sun et al., 2025b; Pun et al., 2025; Gumin et al., 2025; Gu et al., 2025; Dong et al., 2025; Abdelreheem et al., 2025). Some of these methods leverage priors from generated images to construct 3D scenes (Wang et al., 2024b; Ling et al., 2025; Dong et al., 2025; Gu et al., 2025); however, they often lack fine-grained control and suffer from occlusion issues inherent to 2D image generation. Other approaches directly prompt LLMs or VLMs to predict object placements (Wang et al., 2023b; Sun et al., 2025b; Abdelreheem et al., 2025), but this requires strong 3D spatial and geometric reasoning capabilities, which remain a challenge for current models. A third strategy uses LLMs to generate spatial predicates that are then resolved by external solvers (Wen et al., 2023; Yang et al., 2024c; Gumin et al., 2025; Pun et al., 2025). Nevertheless, these methods typically (1) rely on simplified collision avoidance mechanisms (e.g., axis-aligned bounding boxes) that operate primarily

in 2D, and (2) lack feedback or self-correction loops, thereby limiting their scalability in crowded scenes and their capacity to capture the natural complexity of real-world placements. Finally, existing approaches fail to account for the rich physical interactions found in real-world 3D scenes, such as stacking, containment, and support relationships, along with the detailed physical properties of objects. These aspects are essential for generating physically plausible layouts. Although some works (Wang et al., 2024b; Ling et al., 2025; Sun et al., 2025b) can produce visually convincing stacking, their results do not guarantee physical accuracy, often leading to object penetrations or unstable configurations.

To address these challenges, we propose an agent-based framework, coupled with a physics engine, that generates physically accurate scenes while also taking objects' physical properties into consideration. Specifically, our system consists of three main components: (1) an LLM agent that takes a scene description as input and proposes a set of objects along with spatial and physical predicates, (2) a solver equipped with a physics engine that realizes these predicates into concrete scene parameters, and (3) a feedback system that analyzes the generated scene and provides corrective signals to the LLM agent for refinement or further generation.

Our framework offers several benefits. First, the core of the framework is a physics simulator with heuristic methods that generates scenes with a high degree of naturalness and physical plausibility, as well as complex stacking behaviors (Fig. 1a). By allowing objects to interact and settle under simulated physical forces, our method can produce intricate and realistic placements. It also leverages probabilistic programming techniques (Wang et al., 2024a) to measure the stability of the placement, allowing optimization towards user-intended corner cases, such as unstable stacks (Fig. 1a.1) or partially supported placements (Fig. 1a.2). Second, the LLM-powered agentic framework allows for strong controllability and text-following ability, capable of following highly detailed instructions (Fig. 1b.1) or adjusting properties such as the compactness and messy level of a scene (Fig. 1b.2) Third, our feedback system enhances the agent's ability to perceive and improve scenes through multiple modalities: grammar checks, failure reason detection, empty-space identification, VQA-based metrics (Lin et al., 2024) for evaluating clutter and organization, and stability assessments from the physics engine and probabilistic programming. Together, these components enable efficient self-correction and fine-grained control over scene arrangement.

Besides the ability to generate physically accurate and structurally complex scenarios for robot manipulation, our approach can generate diverse scenes free from training data requirements. As illustrated in Fig. 1c and Fig. 5, it can produce diverse environments across settings such as boxes, kitchen setups, shelves, tabletops, and floor arrangements. Experimental results show that our method outperforms prior approaches in visual quality, semantic correctness, and physical accuracy. Furthermore, we demonstrate its utility for robotic learning by automatically collecting demonstration data in generated scenarios and training a policy that successfully transfers to unseen, human-designed setups—highlighting the potential of our framework for automatic data generation in embodied AI.

We summarize our main contributions as follows:

- We propose PhyScensis, an agentic framework that leverages procedural predicates to generate interactive physical scenes, together with a comprehensive feedback system that enables the agent to perceive its environment more effectively and iteratively refine scene arrangement.
- We incorporate physics simulation into the arrangement process, ensuring natural object placement, rich stacking behaviors, and high physical plausibility.
- We demonstrate that our framework significantly outperforms prior methods in generating complex, physically-plausible scenes. Through quantitative and qualitative experiments, we validate that our approach produces arrangements with a level of intricacy and naturalness that is unattainable by models lacking a physics-based generative process.

## 2 RELATED WORKS

### 2.1 INTERACTIVE PHYSICAL SCENE GENERATION

Numerous recent works have studied automatic scene generation for embodied AI, spanning navigation and manipulation applications. One line of research focuses on procedural generation with

manually defined rules (Deitke et al., 2022; Raistrick et al., 2024), which provides precise control but relies on expert-designed heuristics. Another direction trains generative models, such as transformers or diffusion models, on large-scale 3D scene datasets (Paschalidou et al., 2021; Feng et al., 2024; Tang et al., 2023; Yang et al., 2024b; Pfaff et al., 2025), capturing common spatial patterns from data. LLM-based approaches have also emerged, including those that generate symbolic constraints for external solvers (Wen et al., 2023; Yang et al., 2024c; Pun et al., 2025; Gumin et al., 2025), or directly predict placements through prompting (Wang et al., 2023b; Abdelreheem et al., 2025; Sun et al., 2025b). Meanwhile, image-driven pipelines (Lei et al., 2023; Wang et al., 2024b; Ling et al., 2025; Gu et al., 2025; Dong et al., 2025) exploit 2D generative priors such as image generation (Podell et al., 2023; OpenAI, 2025), depth estimation (Yang et al., 2024a; Ke et al., 2024), object recognition (Liu et al., 2024) and segmentation (Kirillov et al., 2023) to construct layouts. Together, these approaches have established diverse strategies for scene synthesis, from rule-based systems to data-driven and LLM-guided frameworks. Our work builds on this landscape by emphasizing richer physical interactions and properties, which remain less explored in prior efforts.

## 2.2 PHYSICALLY ACCURATE GENERATION

There is a growing body of work on incorporating physics into 3D modeling. Some approaches leverage video input, where temporal context facilitates the inference of physical properties such as material parameters (Zhong et al., 2024) and geometry (Li et al., 2022). Others follow a two-stage pipeline reconstructing object geometry from multi-view images, then applying physical simulations to the recovered shape (Feng et al., 2023; Xie et al., 2023; Mezghanni et al., 2021b;a). Another direction attempts to infer physical properties directly from static images using data-driven models to estimate attributes like shading, mass, and material (Zhai et al., 2024; Bell et al., 2014; Standley et al., 2017). Guo et al. (2024) further introduces the notion of physical compatibility for single-object modeling from a single image. In contrast, our work targets physically accurate generation for the 3D scene, which involves multi-object interaction and global contact constraints, making the problem inherently more challenging.

## 3 METHOD

Starting from an empty state with a supporting surface, our method generates complex physical scene layouts from a natural language prompt. The generated scene configuration includes both the 3D placement of assets and their associated physical properties. As illustrated in Fig. 2, our pipeline consists of three stages: (a) an LLM agent takes a user prompt as input and generates a set of spatial and physical predicates, together with object descriptions for retrieval; (b) a solver computes the final scene configuration using a sample-based constraint solver for spatial predicates and a physics engine for physical predicates; (c) a feedback system reports whether the scene was successfully solved or indicates possible causes of failure, which enables the LLM agent to iteratively refine and regenerate predicates.

### 3.1 ASSET DATASET

We build our 3D asset dataset with BlenderKit (BlenderKit Contributors, 2025), using Chat-GPT (Hurst et al., 2024) to annotate the front direction, text description, supporting probability, and ranges of physical properties such as mass, friction, and center-of-mass shift. During scene generation, if there is no match for a given object description, we employ a text-to-3D pipeline to generate new assets. More details are in Appendix A.4.1.

### 3.2 PREDICATE DEFINITION

We define a set of spatial and physical predicates that govern the placement and orientation of objects in a scene. Spatial predicates specify 2D positional or rotational relations in the $x$–$y$ plane, while physical predicates capture more complex 3D interactions such as stacking, supporting, or containment. These predicates are resolved by the solver to update object positions and orientations consistently. Each predicate may also include parameters (e.g., distance, alignment offsets), which are initialized by the agent and optimized by the solver. The complete definitions and formatting conventions for each predicate are provided in Appendix A.4.2.

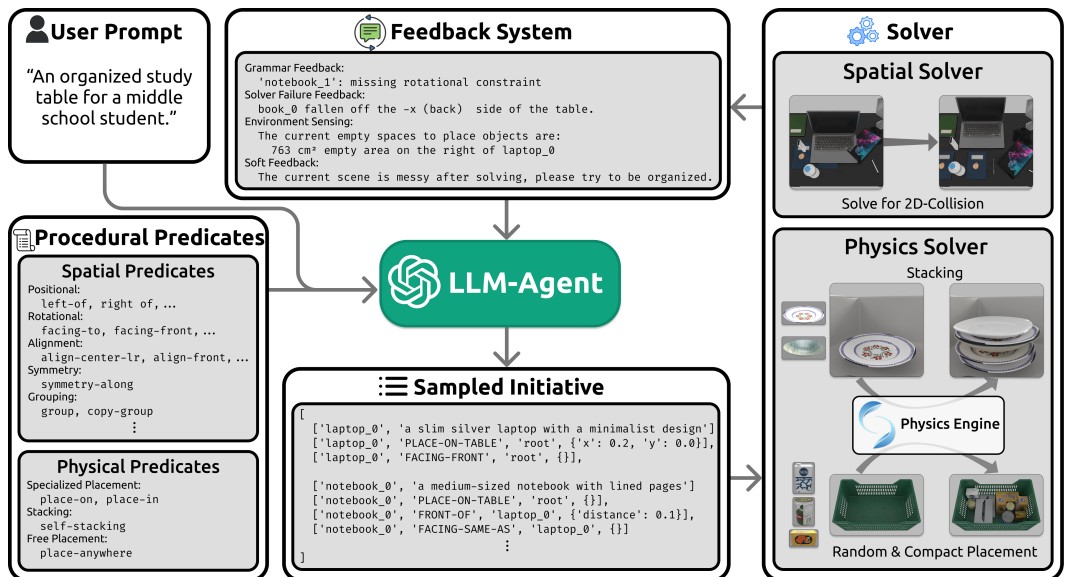

Figure 2: Our framework consists of three components: (a) an LLM agent that takes a user prompt and generates spatial and physical predicates, along with object descriptions for retrieval; (b) a solver that computes the final scene using a physics engine for physical predicates and a sample-based constraint solver for spatial predicates; and (c) a feedback system that reports success or diagnoses failure, allowing the LLM agent to iteratively refine and regenerate predicates.

**Spatial Predicates.** Spatial predicates define object placement in the $x$–$y$ plane and include relative positioning, alignment, symmetry, orientation, and grouping:

- **2D Positional:** `left/right/front/back-of` places one object relative to another along the $x$ or $y$ axis with a specified distance. `place-on-base` places an object on the supporting surface with specified or randomized $x$ and $y$ coordinates.
- **Alignment:** `align-left/right/front/back`, `align-center-lr/fb` constrain bounding box edges or centers.
- **Rotation:** `facing-left/right/front/back`, `facing-to`, `facing-same-as`, `facing-opposite-to`, `orient-by-relative-side`, and `random-rot` determine an object's yaw orientation relative to another object or the global coordinate system.
- **Symmetry:** `symmetry-along` places an object symmetrically with respect to a reference object and axis.
- **Grouping:** `group` creates a virtual group of objects with a defined anchor, while `copy-group` instantiates a new group by duplicating an existing one and preserving its relative structure.

**Physical Predicates.** Physical predicates, as shown in the examples in Fig. 3, capture 3D interactions, including supporting, containment, and stacking:

- **Container Placement:** `place-in` handles placing a batch of objects into a container.
- **Stacking:** `place-on` places a new object on an existing object with physical plausibility, allowing the user to specify the relative position, support ratio (the ratio of the contact area to the object's bottom area), and stability.
- **Free Placement:** `place-anywhere` assigns a random, supported, and penetration-free position when explicit constraints are unnecessary.

These predicates enable both fine-grained spatial reasoning and the generation of physically plausible arrangements. Spatial constraints ensure consistent 2D positioning and orientation, while physical predicates capture richer 3D relationships such as support, stacking, and containment.

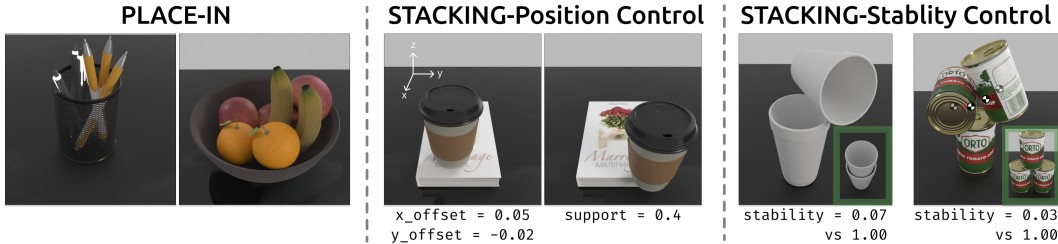

Figure 3: Examples of placements generated by physical solvers.

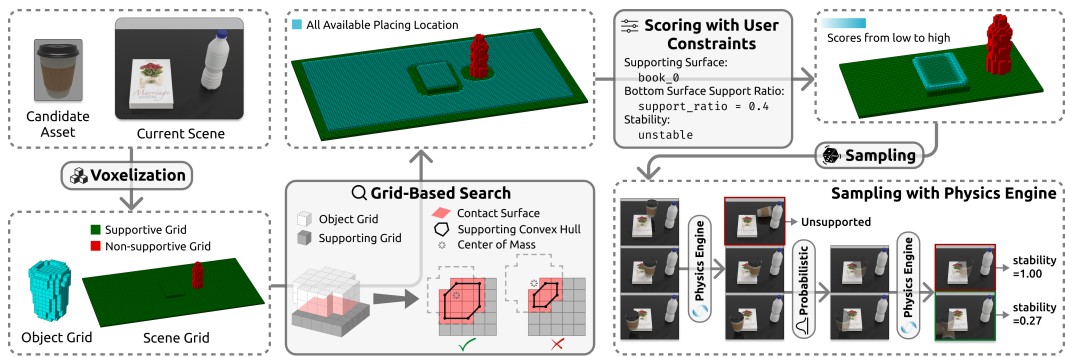

Figure 4: The stacking generation pipeline uses an occupancy-grid-based heuristic to efficiently compute candidate placement locations via grid search, which are then ranked by user requirements. A physics simulator verifies physical validity (e.g., whether an object will fall), and probabilistic programming further assesses stability, enabling control over the robustness of valid states.

## 3.3 SOLVER

Our solver consists of two components: a *spatial solver* and a *physical solver*. The spatial solver resolves spatial predicates to determine 2D positions and orientations of objects, while the physical solver addresses physical predicates to construct complex supporting and stacking behaviors, with the help of a physics engine. In practice, we first apply the spatial solver to determine the 2D placements for objects specified to be placed on the supporting surface (such as a table), and then apply the physical solver to solve for objects with 3D placements.

**Spatial Solver.** The spatial solver focuses on objects defined by spatial predicates that can be resolved deterministically in the $x$–$y$ plane, which includes all objects placed on the supporting surface. The spatial solver first determines whether an object is *fully solved*, which means its x-coordinate, y-coordinate, and yaw orientation are determined by the predicates' parameters or can be inferred from the predicates. If the object is *fully solved*, an initial position and orientation can be assigned to it as a candidate placement. Otherwise, the feedback system alerts the LLM agent, prompting it to provide additional predicates.

To evaluate candidate placements, we compute the 2D convex hull of each asset and use both (i) convex-hull overlap area between objects and (ii) the distance between an object's center and the table boundary as penalty terms. Compared to axis-aligned bounding boxes, convex-hull overlap provides more precise collision checks while remaining significantly faster than full 3D mesh intersection tests, thereby enabling a richer set of feasible configurations.

Inspired by prior optimization methods (Gumin et al., 2025), we iteratively refine the parameter set of all spatial predicates by optimizing one parameter at a time. If the resulting penalty falls below a predefined threshold, the placement is accepted as penetration-free and within the table boundaries. Otherwise, if the solver fails to converge within a fixed number of steps, the case is reported to the feedback system as unsolvable under the given predicate set.

**Physical Solver.** Once spatial placements are established, we resolve physical predicates using the physical solver. Different types of physical predicates are handled as follows:

For `place-in`, we adopt a physics-based packing strategy similar to Blender's physics placer Bal (2024): objects are initialized above the target container and released with forces to settle into penetration-free, physically plausible poses.

For `place-on` and `place-anywhere`, we adopt an occupancy-grid-based heuristic combined with a physics engine (Fig. 4). Both the current scene and the candidate asset are voxelized into occupancy grids, and feasible placements are identified as grid positions that are penetration-free and whose projected center of mass lies within the supporting convex hull. We then score the filtered candidates based on predicate parameters. For example, in Fig. 4, the predicate specifies placing a cup on a book with a bottom-area support ratio of 0.4 and aims for an unstable placement, so candidates near the book's edges receive higher scores. While this heuristic provides efficient priors, it is limited by grid resolution and the lack of continuous physics. To ensure physical plausibility, we subsequently validate sampled candidate placements using a physics engine, where only objects with no large displacement after the simulation are regarded as successfully placed. More details are available in Appendix A.4.3.

Moreover, we employ probabilistic programming techniques (Wang et al., 2024a) to measure the *stability* of a placement by sampling perturbations around it. A placement is considered more stable if nearby perturbations also result in valid, balanced states. Specifically, we sample the 3D position, Euler angles, mass, center-of-mass shift, and friction coefficient around the current state using a normal distribution within the range of that object. This allows us to compute the probability of each state being sampled. We then run simulations in the physics engine to determine whether the system remains stable or falls, and finally estimate the overall stability probability of the state using a Bayesian approach. More details are explained in Appendix A.4.4. With this framework, we can choose to further repeatedly optimize those parameters to a more unstable state by choosing the unstable yet still not falling configurations at each iteration, achieving the extremely unstable placements as in Fig. 3.

### 3.4 FEEDBACK SYSTEM

After processing with the solver, we provide feedback to the LLM agent to close the loop and enable iterative refinement of the scene. The feedback serves three primary purposes: detecting errors, diagnosing unsolvable cases, and evaluating successfully generated scenes.

**Grammar Feedback.** First, we check whether the generated predicate set is grammatically valid and fully parsable. This includes verifying that each object is *fully solved* according to the criteria defined earlier. If any predicate is ill-formed or an object remains unsolved, the system returns explicit feedback identifying which objects and predicates are problematic.

**Solver Failure Feedback.** If the grammar is correct but the solver fails to find a valid configuration, we provide diagnostic feedback describing which objects are invalid and why. Possible failure modes include object penetrations, falling outside the supporting surface, or inability to find a feasible stacking pose. In addition, we estimate the crowding level of the scene and heuristically identify empty areas. This information is then communicated to the agent in natural language, e.g., "There is an empty region behind the laptop on the left side of the table.". Such feedback helps the agent adjust its predicates and resample placements in less congested regions.

**Success Feedback.** When the solver successfully produces a valid scene, we return evaluation metrics to guide further refinement. These include: (1) **Stability:** a stability score estimated by the physics engine together with the probabilistic programming techniques; (2) **Visual Quality:** a VQA score assessing whether the scene appears organized, cluttered, or messy; and (3) **Heuristic Measures:** additional statistics such as surface coverage, compactness, and the number of objects placed. The agent incorporates these measurements to decide whether the current scene is satisfactory or whether further adjustments and resampling are necessary.

## 4 EXPERIMENTS

To validate the effectiveness of PhyScensis, we compare it with state-of-the-art baselines, both qualitatively and quantitatively. We also conduct a robotic experiment to validate the effectiveness of our generated scenes for robot manipulation policy training. In ablation study, we evaluate the effectiveness of the feedback system within our agentic framework, as well as our placement design. We further present an analysis on stability control in Appendix A.5.4.

### 4.1 BASELINES

We compare our method with open-vocabulary scene generation approaches capable of placing objects in local scenarios. Specifically, we consider **3D-Generalist** (Sun et al., 2025b), which uses Molmo (Deitke et al., 2025) to iteratively point to 2D pixels for object placement, and **Architect** (Wang et al., 2024b), which leverages image inpainting to generate placements, followed by recognition, segmentation, and depth estimation to infer 3D positions. Both baselines require asset retrieval; to ensure a fair comparison, we use the same asset dataset as our method. Further implementation details are provided in Appendix A.5.1. Additionally, in Appendix A.3 we compare our work with **LayoutVLM** (Sun et al., 2025a), which is the state-of-the-art room level scene generation method, and **ClutterGen** (Jia & Chen, 2024), which is dedicated to generating cluttered object arrangements on a supporting surface.

| Method | Metrics | | | Reaching ↑ | Placing ↑ |
|---|---|---|---|---|---|
| | VQA Score ↑ | GPT Ranking ↓ | Settle Distance ↓ | | |
| Architect | $0.493 \pm 0.392$ | $2.607 \pm 0.673$ | $0.405 \pm 0.471$ | 3/10 | 0/10 |
| 3D-Generalist | $0.578 \pm 0.399$ | $1.946 \pm 0.731$ | $0.033 \pm 0.048$ | 4/10 | 1/10 |
| Ours | $\mathbf{0.704 \pm 0.425}$ | $\mathbf{1.429 \pm 0.562}$ | $\mathbf{0.003 \pm 0.008}$ | **9/10** | **3/10** |

Table 1: Quantitative comparison of PhyScensis with baselines.

**Metrics.** We evaluate generated scenes using the following metrics: **VQA Score** (Lin et al., 2024): a VQA model estimates the probability that the rendered image matches the input caption; **GPT Ranking**: GPT ranks rendered scenes for a given caption, and we report the average rank as a score; **Settle Distance**: after simulating for a fixed number of timesteps, we compute the average displacement of objects, indicating physical stability.

**Analysis.** As shown in Table 1, our method outperforms both baselines across all metrics. For **3D-Generalist** (Sun et al., 2025b), scene quality is limited by the VLM's relatively weak spatial reasoning ability. The VLM is capable of simple pointing tasks, such as identifying an object's position, but struggles to effectively reason about a suitable placement for a given object. For **Architect** (Wang et al., 2024b), performance depends heavily on the quality of inpainted images produced by Stable Diffusion XL (Rombach et al., 2021). However, the generated images are often imperfect, resulting in suboptimal placements and lower quality scores.

Neither baseline adequately addresses physical accuracy. **3D-Generalist** applies basic collision avoidance but does not enforce stability. As a result, it struggles with complex stacking scenarios (e.g., placing chopsticks on a bowl, as shown in Fig. 1). **Architect** relies solely on depth estimation, frequently causing inter-object penetrations, which might cause an explosion of the simulation. These shortcomings are reflected in the higher settle distance reported in Table 1. Furthermore, both baselines depend strongly on 2D visualizations, making them unreliable for cluttered scenes with significant occlusions. **Architect** performs only a single inpainting step for local scenarios, while **3D-Generalist** requires repeated queries for each placement, limiting its scalability. As illustrated in Fig. 5, our method consistently produces more complex and cluttered scenes, with the ability to stack objects and iteratively expand scene complexity.

### 4.2 ABLATION STUDY

We conduct an ablation study to evaluate the contribution of our feedback system in correcting failure cases. We measure the average number of resampling attempts required for self-correction

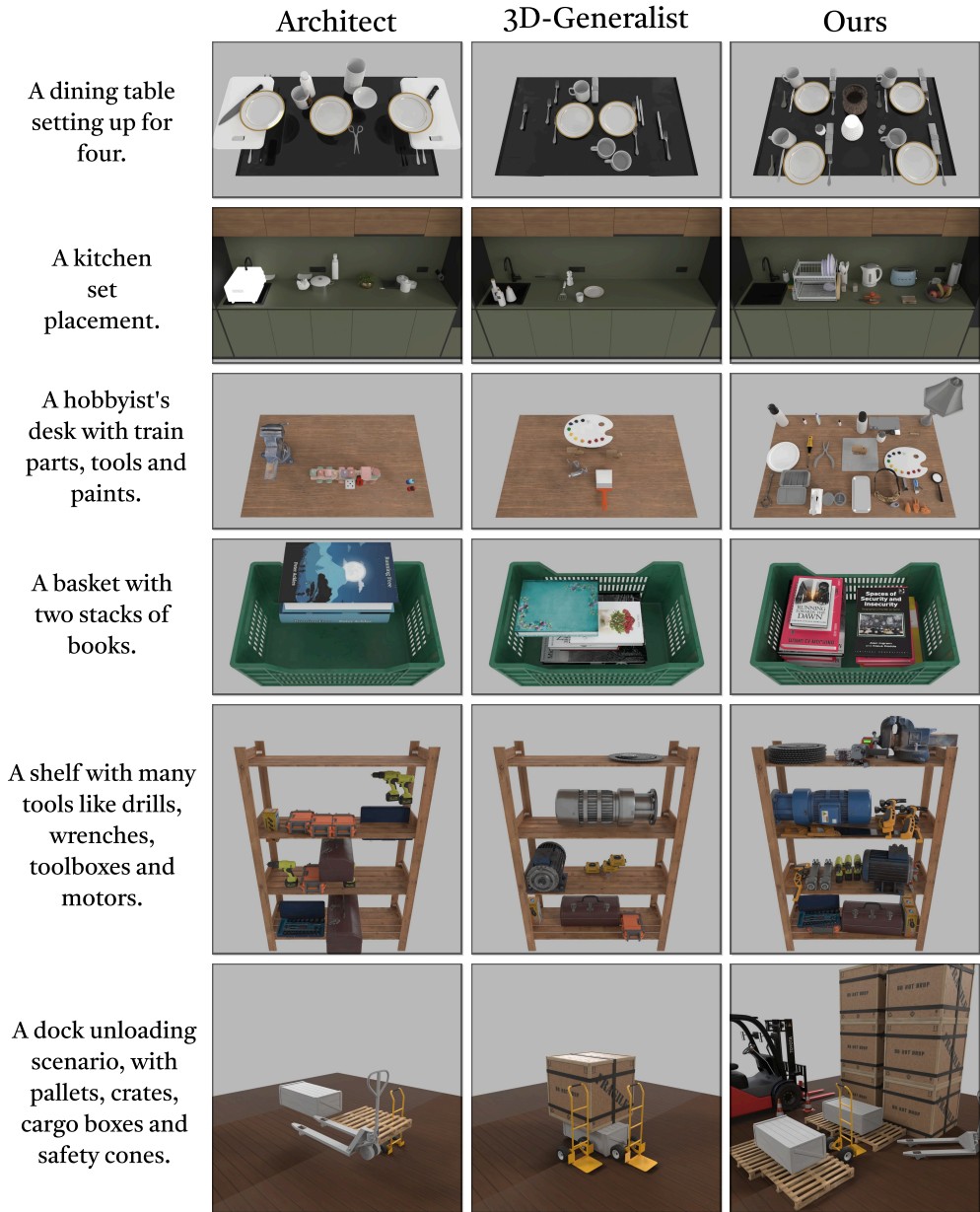

Figure 5: Qualitative comparison of PhyScensis with baselines for different generating scenarios.

| Method | Metrics | |
| --- | --- | --- |
| | Retry Times ↓ | Time-cost ↓ |
| **Ours w/o feedback** | $1.69 \pm 1.92$ | $132.29 \pm 78.38$ |
| **Ours w/o report** | $1.43 \pm 1.55$ | $126.09 \pm 59.19$ |
| **Ours Visual** | $\mathbf{0.95 \pm 0.91}$ | $120.65 \pm 53.62$ |
| **Ours** | $1.04 \pm 1.41$ | $\mathbf{106.41 \pm 55.53}$ |

Table 2: Comparison with ablated versions on time-cost and retry times.

| Method | Metrics | | |
| --- | --- | --- | --- |
| | VQA Score ↑ | GPT Ranking ↓ | Settle Distance ↓ |
| **Random** | $0.415 \pm 0.363$ | $2.706 \pm 0.666$ | $0.004 \pm 0.003$ |
| **LLM-Only** | $0.592 \pm 0.401$ | $1.882 \pm 0.676$ | $0.154 \pm 0.133$ |
| **Ours** | $\mathbf{0.704 \pm 0.425}$ | $\mathbf{1.411 \pm 0.492}$ | $\mathbf{0.003 \pm 0.008}$ |

Table 3: Comparison with ablated versions on scene quality.

and compare the following variants: **Ours w/o feedback**: Removes the feedback system entirely, providing only a binary success/failure signal; **Ours w/o report**: Retains most feedback but excludes the reporting of empty regions; **Ours Visual**: Augments the full framework with visual feedback

for failure correction. The quantitative results are presented in Table 2. A detailed analysis of these results is deferred to Appendix A.5.2.

To evaluate the contribution and effectiveness of our designed predicates and solvers, we conduct another ablation to compare our full method against two baselines: **Random**: After an LLM proposes a list of objects, we place them at random, collision-free locations and orientations on the table using our spatial solver. **LLM-Only**: An LLM directly proposes objects along with their specific locations and orientations, similar to LayoutGPT (Feng et al., 2024). We use the same metrics as in Sec. 4.1: VQA Scores, GPT Ranking, and Settle Distance. The results are summarized in Table 3, with a full analysis available in Appendix A.5.3.

## 4.3 ROBOT EXPERIMENT

To evaluate the usefulness of generated scenes for robotics, we conduct an imitation learning experiment. We constrain the scene distribution with the prompt *"a dining table set up for four people"* and benchmark the task: *"pick up the leftmost cup and place it on the rightmost plate."* For simplicity, we fix the cup and plate assets across scenes. We collect 300 scenarios for each method (ours and the baselines) and record one demonstration trajectory per scenario, resulting in training datasets for a diffusion policy (Chi et al., 2024). For evaluation, we additionally collect 10 scenarios designed by humans and measure two metrics: *reaching success rate* (the robot arm reaches the correct cup) and *placing success rate* (the full trajectory succeeds).

As shown in Fig. 11 in Appendix, policies trained on our generated data generalize effectively to unseen human-designed scenes. Quantitative results in Tab. 1 further demonstrate that our method achieves higher success rates than the baselines, together with Fig. 12, indicating that our generated scenes better approximate real-world distributions.

## 4.4 USER STUDY

We conducted a user study comparing our method against two main baselines: Architect (Wang et al., 2024b) and 3D-Generalist (Sun et al., 2025b). Specifically, we selected 6 prompts and randomly sampled one result for each of the three methods, creating a total of 18 evaluation cases. For each case, we asked users to rate the generated scene on a scale of 1–5 across three dimensions: text alignment, naturalness & physical plausibility, and complexity. We collected responses from 20 participants; the results are shown in Table 4.

| Baseline | Match with Text ↑ | Naturalness & Physics ↑ | Complexity ↑ |
|---|---|---|---|
| Architect | 2.68 | 2.65 | 2.69 |
| 3D-Generalist | 2.54 | 2.72 | 3.04 |
| **Ours** | **4.04** | **3.98** | **3.82** |

Table 4: User Study Results. We report the mean scores from user evaluations (scale 1-5).

Our method significantly outperforms the other two baselines, which is consistent with our quantitative metrics. This result also aligns with our qualitative observations: Architect estimates object layout directly from inpainted images, often leading to object penetrations and implausible physics. Meanwhile, 3D-Generalist employs a VLM to output placement positions in pixel space. However, the VLM's limited spatial reasoning capability fails to yield reasonable layouts for complex prompts, resulting in this baseline receiving the lowest scores for text alignment.

## 5 CONCLUSION

In this work, we tackle the task of generating complex 3D scenes with rich physical interactions, an area that has not been thoroughly explored in prior work. To address this challenge, we propose PhyScensis, an LLM-based agentic framework augmented by a physics engine. The LLM agent proposes a list of assets and placement predicates based on a user's description, ensuring strong controllability, while the solver powered by a physics engine ensures natural and physically accurate placements. A feedback system further enables scene refinement and iterative generation. Experimental results show that our framework can generate diverse physical scenes with complex stacking behaviors and natural placements, outperforming prior work both qualitatively and quantitatively.

ACKNOWLEDGEMENT

We thank the anonymous reviewers for their helpful suggestions. This work was supported in part by MURI grant N000142412748 and NSF grants IIS-2441250 and IIS-2404386. This work was partially completed while Yian was interning at Genesis AI.

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

## A  APPENDIX

### A.1  THE USE OF LARGE LANGUAGE MODELS

In the preparation of this manuscript, we utilized LLM as a proofreading tool. Its application was strictly limited to checking for grammatical errors, spelling mistakes, and improving sentence clarity. The LLM did not contribute to any scientific ideas, experimental results, or the core structure of the paper.

### A.2  MORE QUALITATIVE RESULTS AND FAILURE ANALYSIS

We present more qualitative results on the anonymous website https://physcensis.github.io, including generated scenes, visualization videos, and examples of our synthesized manipulation data.

#### A.2.1  QUALITATIVE COMPARISON WITH ABLATION BASELINES

We show qualitative comparisons with our ablation baselines: Random and LLM-Only. The Random baseline places objects at random positions on the table, while the LLM-Only baseline uses an LLM to directly output object positions and orientations. Both baselines use a simple collision solver to ensure collision-free placement. From the qualitative results in Fig. 6, although the LLM-Only baseline can produce reasonable layouts, its generated scenes lack complex stacking or containment behaviors, making them look less natural and diverse.

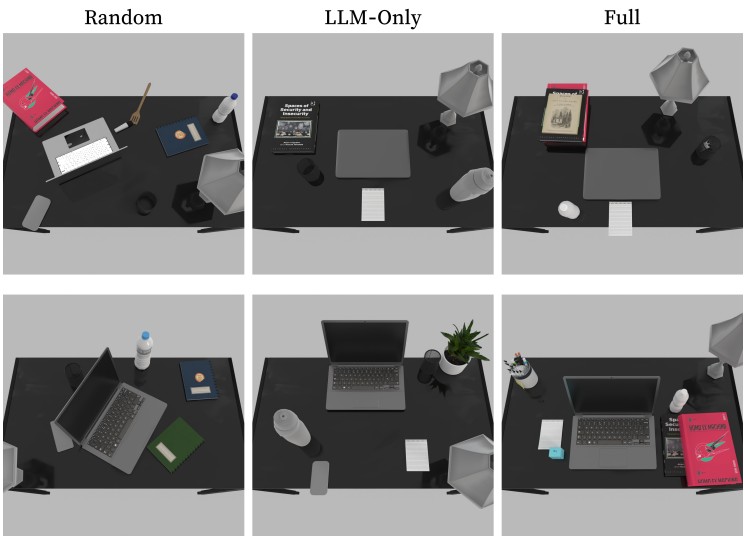

Figure 6: Qualitative comparison with Random and LLM-Only baselines.

#### A.2.2  FAILURE CASES WITH BAD RETRIEVAL

One major factor contributing to poor generation quality is inaccurate retrieval. Although our pipeline checks the similarity between the retrieved object and the text description to filter out mismatched assets, it fails in some corner cases. For example, in the first image of Fig. 7, the two plastic bottles belong to a single asset that aligns well with the description. Consequently, it is successfully retrieved; however, after optimization, this results in a floating bottle. A similar issue occurs in the second example in Fig. 7. Such failure cases could be mitigated by using a more comprehensive filtering pipeline when constructing the asset library for retrieval.

#### A.2.3  FAILURE CASES OF SOLUTION NOT FOUND

During our iterative scene generation process, as the scene becomes populated with many objects, the spatial solver may fail to find a collision-free solution for a new set of objects. Two such cases,

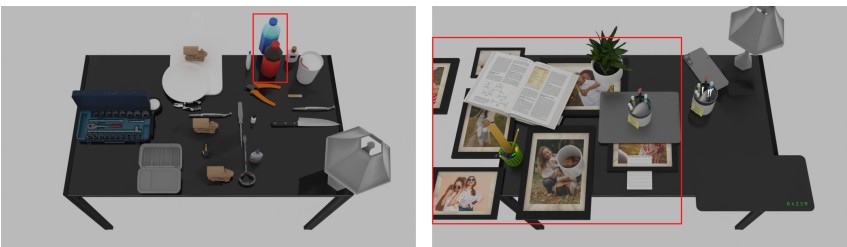

Figure 7: Examples of cases when the retrieved objects are not reasonable.

where valid placements could not be found, are shown in Fig. 8. It's common when the table is already covered with a lot of assets. Such failure cases could be mitigated by explicitly encouraging the agent to use physics predicates—for example, to stack objects or to use PLACE-ANYWHERE to automatically find valid placements and further enrich the scene.

Before            After

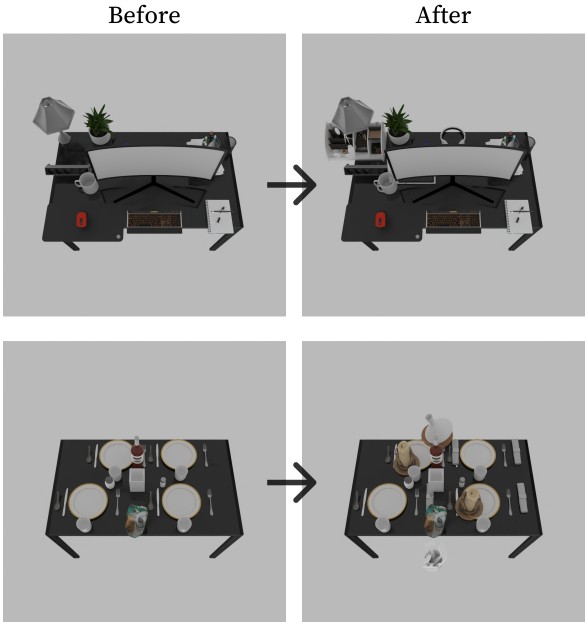

Figure 8: Examples of cases when spatial solver cannot find a solution.

### A.2.4 FAILURE CASES WITH UNSTABLE STACKING

When processing physical predicates to solve for complex stacking, the scene could sometimes be unstable when too many objects are added. In this case, objects will fall wherever the new objects placed, so that no new objects can be added, even though there is enough space during the grid-based search. For example, in Fig. 9, the stack of plates leads to an unstable simulation where future stack always falls. In such cases, our pipeline will fail to stack new assets within the time limit.

### A.2.5 FAILURE CASES WITH LOW VQA SCORE

In Fig. 10, we show some of our generated results that receive low VQA scores. Based on these results, we observe that the VQA score can, to some extent, reflect the degree of alignment between the text prompt and the generated scene. For example, the first example may have received a low score because there are no snacks in the bowl; the second, because the laptop and display do not match the profile of a middle school student; and the third, because the scene is not messy enough.

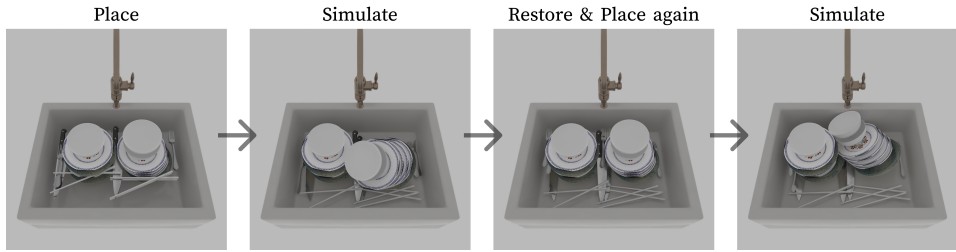

Figure 9: Examples of unstable stack.

However, in terms of scale, the VQA score does not always reflect human perception. For instance, while the third example is indeed not messy enough, a near-zero score is disproportionately low.

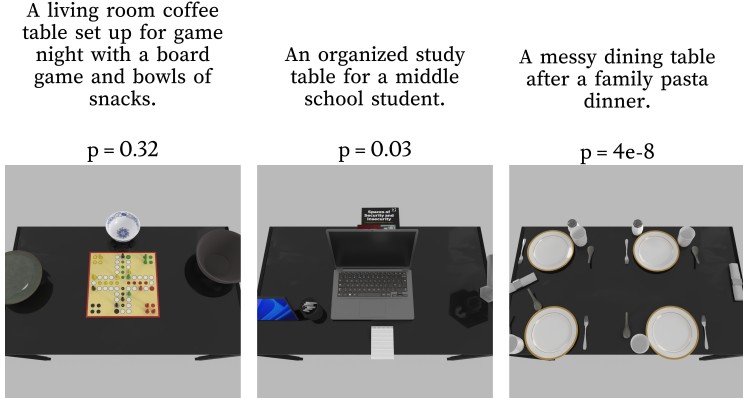

Figure 10: Examples of results with low VQA Scores.

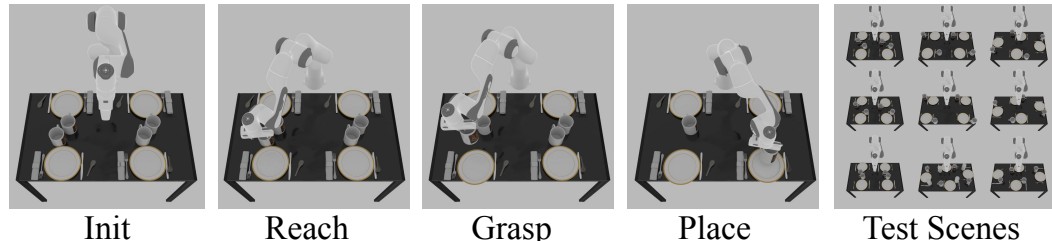

Figure 11: An example of policy rollout.

### A.3 ADDITIONAL BASELINE COMPARISON

#### A.3.1 LAYOUTVLM

LayoutVLM (Sun et al., 2025a) adopts a similar method for 3D scene-level layout generation, using LLMs to generate proposals for object positions and relationships, followed by a differentiable solver to determine the final layouts. We present additional quantitative results (Table 5) and qualitative results (Fig. 13). Note that LayoutVLM requires a text prompt paired with a list of specified 3D assets as input. To adapt this method to our setting of text-only conditioned scene generation, we introduce an additional pipeline that queries GPT-4o to generate a list of object descriptions, and then uses CLIP features to retrieve corresponding 3D assets from the database. To ensure a fair comparison, we process our 3D asset database (also sourced from Objaverse) into the format required by LayoutVLM.

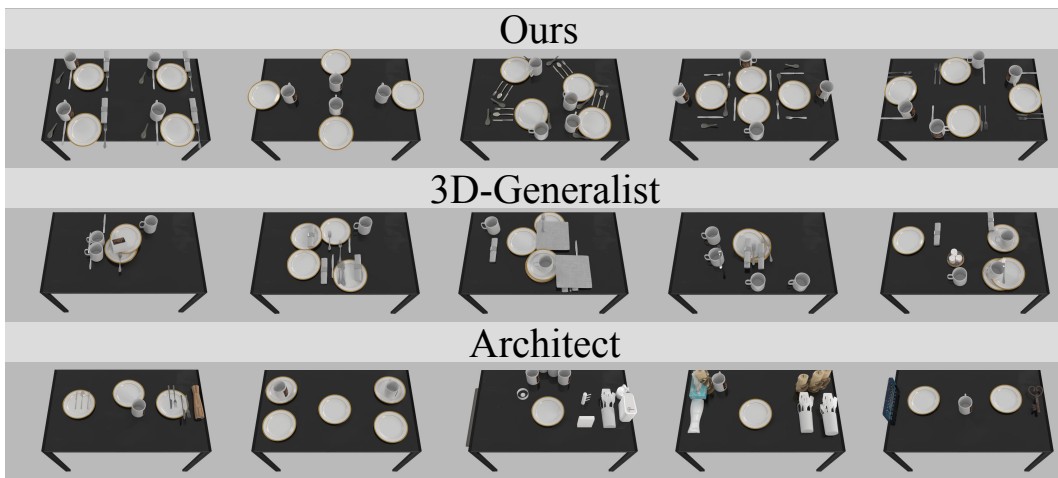

Figure 12: Visualized samples from generated scene for the robot manipulation task. The result shows that our method, while still being diverse, resembles more to the human intuitions.

| Method | Metrics | | |
| --- | --- | --- | --- |
| | VQA Score ↑ | GPT Ranking ↓ | Settle Distance ↓ |
| Architect | $0.493 \pm 0.392$ | $3.250 \pm 0.947$ | $0.405 \pm 0.471$ |
| 3D-Generalist | $0.578 \pm 0.399$ | $2.393 \pm 0.899$ | $0.033 \pm 0.048$ |
| LayoutVLM | $0.648 \pm 0.446$ | $2.643 \pm 1.109$ | $0.115 \pm 0.238$ |
| Ours | $\mathbf{0.704 \pm 0.425}$ | $\mathbf{1.714 \pm 0.920}$ | $\mathbf{0.003 \pm 0.008}$ |

Table 5: Additional quantitative comparison of PhyScensis with baselines, including LayoutVLM.

Our method outperforms LayoutVLM both qualitatively and quantitatively. A primary limitation of LayoutVLM is its design focus on room-level layout generation. In such settings, the environment is spacious relative to the furniture, allowing for a large optimization search space. However, in our experiments, we constrain the optimization to a small area (as all objects must be placed on a tabletop). Under these conditions, LayoutVLM's optimization often fails when the object count is high, such as in the prompt "A stack of 5 books on the table". In these cases, the optimization process converges prematurely (e.g., at 50 steps out of 400 iterations). At this stage, physical constraints are not fully resolved, and objects remain cluttered or overlapping, as visible in the kitchen, shelf, and basket scenarios in Fig. 13. This is also the primary reason for the high Settle Distance observed for LayoutVLM in Table 5. Secondly, although LayoutVLM claims to support the stacking of multiple objects, we were unable to generate more complex stacking, such as a stack of books (where assets are different) or more random stacking. In contrast, our method excels at generating these cases with the help of a physics engine. Thirdly, LayoutVLM applies bounding box-based collision checking, similar to most prior works, while we adopt a polygon-based method and a physics engine, which is more precise for crowded scenes with natural object orientations.

### A.3.2 CLUTTERGEN

Previous work, CLUTTERGEN (Jia & Chen, 2024), addresses a related problem of generating complex object arrangements on a supporting surface of a specified size. Their approach employs reinforcement learning to optimize a policy for placing a fixed set of assets, but it does not account for semantic relationships between objects and does not support stacking or containment behavior. In contrast, our method is open-vocabulary, training-free, semantics-aware, and achieves substantially improved performance.

For a fair comparison, we follow the evaluation protocol of CLUTTERGEN, which prioritizes clutterness over semantic correctness. Specifically, we randomly shuffle the asset list and sequentially place objects using the PLACE-ANYWHERE predicate, cycling back to the beginning of the list after

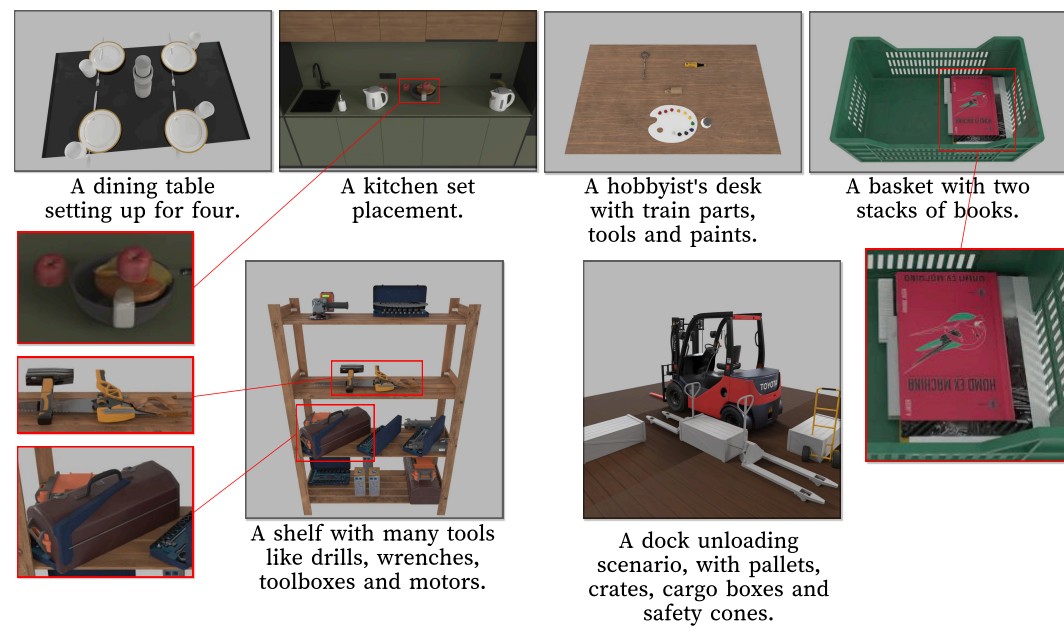

Figure 13: Qualitative results for LayoutVLM, using the same prompt as in Fig 5.

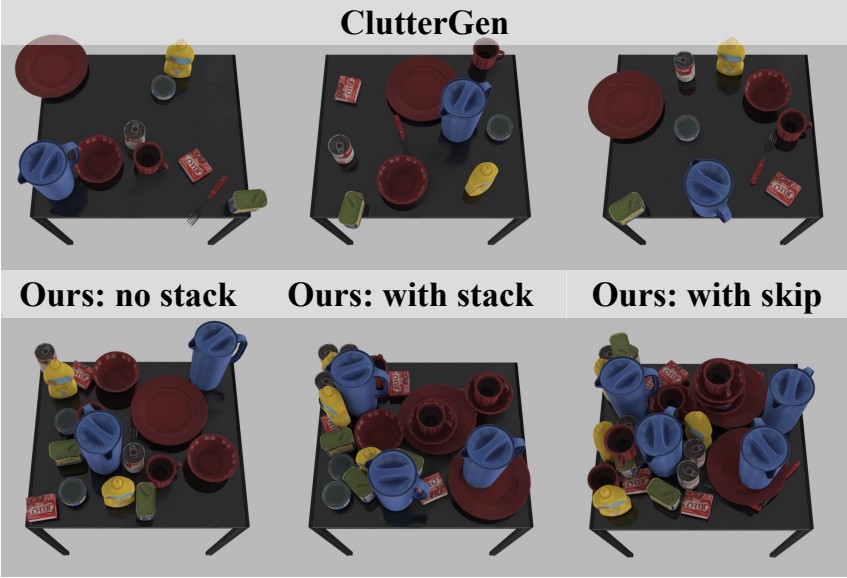

Figure 14: Qualitative comparison between our method and ClutterGen (Jia & Chen, 2024) using the same asset library. We pick the best 3 placements of ClutterGen to render.

exhausting all objects. The process terminates once an object fails to be placed. We use the same set of dining-group assets as CLUTTERGEN and report the total number of objects successfully placed. For CLUTTERGEN implementation, we trained its RL policy for 2.5 million steps before inference with it.

We evaluate our method under three settings: (1) *no-stack*, where stacking is disabled and objects cannot be placed atop one another; (2) *with-stack*, where any physically stable stacking configuration is permitted; and (3) *with-skip*, where objects that fail to be placed are skipped, and the system

continues attempting the remaining objects until no further placements are possible or a time limit is reached.

A qualitative comparison is shown in Fig. 14. Even without stacking, our method produces substantially more compact and cluttered scenes than CLUTTERGEN. Allowing stacking or skipping failed objects further increases scene density. Quantitatively, across 500 sampled scenes, CLUTTERGEN places an average of 3.57 objects, whereas our method places an average of 14.77 and 22.30 objects under the no-stack and with-stack settings, respectively, with the *with-skip* variant achieving even higher counts.

### A.4  IMPLEMENTATION DETAILS

#### A.4.1  DETAILS ON ASSET DATASET PREPARATION

Since our method retrieves 3D assets from an existing dataset rather than generating them from scratch, we pre-process the dataset to support efficient retrieval and provide the necessary annotations.

We build our asset library using BlenderKit (BlenderKit Contributors, 2025), chosen for its high-quality and diverse assets. We construct our asset dataset by downloading 3D models from BlenderKit (BlenderKit Contributors, 2025). To define object categories, we first prompt Chat-GPT (Hurst et al., 2024) to generate a list of common small household objects. For each category, we retrieve the top 30 assets from BlenderKit and annotate them with ChatGPT, resulting in a base dataset of approximately 800 assets.

For each asset, we annotate five key properties. We determine the **front direction** by rendering views from all four sides and asking GPT-4o (Hurst et al., 2024) to identify the correct orientation. We then generate a **text description** of the asset based on its name and front-view image to support semantic retrieval and estimate the asset's **supporting probability**—i.e., how likely it can serve as a support surface (e.g., a closed laptop versus an open one)—as well as ranges of **physical properties** such as mass, friction, and center-of-mass shift. Finally, we compute **embeddings** using both OpenAI text embeddings (OpenAI, 2024) and CLIP (Radford et al., 2021) from the rendered images, enabling multimodal retrieval.

During scene generation, if retrieval fails (i.e., the text–embedding similarity falls below a threshold), we automatically generate a new asset via a text-to-3D pipeline. Specifically, we first generate an image from the description using an image generation model (Rombach et al., 2021) and then call Tripo-AI's image-to-3D API to obtain the corresponding mesh. The new asset is scaled using GPT-estimated dimensions and then passed through the same annotation process before being added to the dataset.

#### A.4.2  DETAILS ON PREDICATE DEFINITIONS

Here we provide the complete prompt provided to the LLM for generating placement predicates for a given scene prompt. Listing 1 presents the system prompt, which details the formatting conventions, mathematical definition, purpose, examples, and additional explanations for each predicate. It also provides notes for the agent, such as naming conventions for new objects, object solving criteria, and other grammatical rules. Listing 2 shows the complete prompting context for predicate generation, including how the scene prompt and feedback are provided to the agent.

```
You are a helpful agent that helps placing a scene.
Your role is to utilize the relationships to construct a whole scene.
    Specifically, you need to give a set of objects, their textual
    descriptions for retrieval, and the relationship between them.
Note that the root node is the supporting surface such as a table. You
    should be careful to keep the object you placed on the support
    surface (inside the range of x, y).
There might be multiple rounds of placement generating and feedbacks, the
     feedbacks might include error messages, or a confirmation with
    request to generate more.

Sections:
```

```
1. Coordinate conventions:
- Root's bbox given in explicitly. When placing objects, ensure any
    coordinate updates keep bbox within these ranges.
- Units: meters.
  Front  = +x
  Back   = -x
  Left   = +y
  Right  = -y

2. RELATIONSHIP DEFINITIONS
When choosing a relationship ["A", predicate, "B", params], apply the
    corresponding update to A's bbox and rotation as defined; ensure A's
    fields are updated before using them in subsequent predicates.

2-1. 2-D SPATIAL:
LEFT-OF
    A.min_y = B.max_y + params["distance"]
RIGHT-OF
    A.max_y = B.min_y - params["distance"]
FRONT-OF
    A.min_x = B.max_x + params["distance"]
BACK-OF
    A.max_x = B.min_x - params["distance"]
Example:
    ["phone_0", "LEFT-OF", "bottle_0", {"distance": 0.1}]

2-2. 2-D ALIGNMENT:
ALIGN-CENTER-LR
    A.center_y = B.center_y
ALIGN-CENTER-FB
    A.center_x = B.center_x
ALIGN-LEFT
    A.max_y = B.max_y
ALIGN-RIGHT
    A.min_y = B.min_y
ALIGN-FRONT
    A.max_x = B.max_x
ALIGN-BACK
    A.min_x = B.min_x
Example:
    ["fork_0", "ALIGN-CENTER-FB", "plate_0", {}]

2-3. 2-D SYMMETRY:
SYMMETRY-ALONG
Purpose: Place object A symmetrically opposite to object B across object
    C
    C = get_info(params["C"])
    A.center_x = 2 * C.center_x - B.center_x
    A.center_y = 2 * C.center_y - B.center_y
Example:
    ["knife_1", "SYMMETRY-ALONG", "knife_0", {"C": plate_0}]

2-4. ROTATION-ONLY:
FACING-TO
    dx = B.center_x - A.center_x
    dy = B.center_y - A.center_y
    A.yaw = math.atan2(dy, dx)
FACING-SAME-AS
    A.yaw = B.yaw
FACING-OPPOSITE-TO
```

```
    A.yaw = (B.yaw + math.pi) % (2*math.pi)
Example:
    ["monitor_0", "FACING-TO", "keyboard_0", {}]

FACING-FRONT
    A.yaw = 0
FACING-BACK
    A.yaw = math.pi
FACING-LEFT
    A.yaw = math.pi / 2
FACING-RIGHT
    A.yaw = - math.pi / 2
Example:
    ["book_0", "FACING-FRONT", "root", {}]

RANDOM-ROT
    A.yaw = random.random() * 2 * np.pi
Example:
    ["pen_0", "RANDOM-ROT", "root", {}]

ORIENT-BY-RELATIVE-SIDE
Purpose: Place the orientation of A relative to B; Orientation depends on
     which side A is relative to B. Like place mouse relative to keyboard
    , utensils relative to plate.
It will automatically gives a compact placement. Good to use together
    with ALIGN-CENTER-LR or ALIGN-CENTER-FB rules.
    def overlapX(A, B)
        return max(0, min(A.max_x, B.max_x) - max(A.min_x, B.min_x))
    def overlapY(A, B)
        return max(0, min(A.max_y, B.max_y) - max(A.min_y, B.min_y))
    A.yaw = default_yaw
    recompute_bbox(A)
    aligned_scale1 = overlapX(A, B) + overlapY(A, B)
    A.yaw = default_yaw + math.pi / 2
    recompute_bbox(A)
    aligned_scale2 = overlapX(A, B) + overlapY(A, B)
    if aligned_scale1 > aligned_scale2:
        A.yaw = default_yaw
    else:
        A.yaw = default_yaw + math.pi / 2
Example:
    ["fork_0", "ORIENT-BY-RELATIVE-SIDE", "plate_0", {}]

2-5. HEIGHT DETERMINATION:
PLACE-ON-BASE
Purpose: Position object A on the root, using the root as the supporting
    surface.
    A.center_x = params["x"]
    A.center_y = params["y"]
    A.min_z = state["root"].max_z
Example:
    ["lamp_0", "PLACE-ON-BASE", "root", { "x": 0.2, "y": 0.3 }]
Other notes:
    You can also set params to empty for PLACE-ON-BASE, so that it will
    refers to other predicates for x and y location

PLACE-ON
Purpose: Position object A on top of object B, using B as the supporting
    surface.
The exact position of A on B will be optimized to ensure physical
    plausibility.
You can also specify some parameters that control the optimization.
    # The position of A relative to B:
    A.center_x = B.center_x + params["x_offset"]
```

```
    A.center_y = B.center_y + params["y_offset"]
    # The overlap ratio of the bottom of A
    params["overlap"] = area(A.bottom.intersect(B.top)) / area(A.bottom)
Example:
    ["notebook_1", "PLACE-ON", "notebook_0", {"x_offset": 0.0, "y_offset
    ": 0.1}]
    or
    ["notebook_1", "PLACE-ON", "notebook_0", {"overlap": 0.8}]
Other notes:
    For this relation, B cannot be "root"; It must be an existing object
    that has a flat surface.
    You can also set params to empty, so that A will be randomly placed
    on B.
    PLACE-ON can be used stack multiple objects on top of each other, e.g
    ., A PLACE-ON B, B PLACE-ON C, C PLACE-ON D, etc.
    When using PLACE-ON, do not use other spatial predicates in 2-D
    SPATIAL or 2-D ALIGNMENT for A, since its position is already
    determined by B, the offsets and the overlap ratio.
    PLACE-ON is processed after other objects except PLACE-ANYWHERE. So
    predicates related to the PLACE-ON object should come after other
    predicates but before PLACE-ANYWHERE.

2-6. GROUPING:
GROUP
Purpose: Define a new group object that aggregates a list of existing
    objects.
    Usage: [group_name, "GROUP", [object_id_1, object_id_2, ...], {"
    anchor": object_id_k}]
    Conventions:
        group_name must be a unique identifier starting with "group_".
        The list must refer only to objects that have already been
    introduced.
    Effect: Creates a virtual/grouped entity "group_name" that represents
     the set of specified objects. The facing direction of the group will
     be defined as the facing direction of the anchor object defined by
    the parameter.
Example:
    ["group_dining_set_0", "GROUP", ["plate_0", "fork_0", "knife_0", "
    cup_0"], {"anchor": "plate_0"}]
    This creates a group named group_dining_set_0 comprising the existing
     objects plate_0, fork_0, knife_0, and cup_0. You don't need to do
    other things since they've already been placed.

COPY-GROUP
Purpose: Instantiate a new group by copying the structure and relative
    positions of an existing group.
    Usage: [new_group_name, "COPY-GROUP", existing_group_name, {}]
    Conventions:
        new_group_name must be a fresh identifier starting with "group_".
        existing_group_name must refer to a group already defined.
    Effect: Creates a new set of objects (with new identifiers) arranged
    in the same relative configuration as in existing_group_name. This is
     useful for duplicating a table arrangement (e.g., another place
    setting).
    Follow-up: After copying, apply spatial predicates to position the
    copied group in the scene. For example:
Example:
    [
        ["group_dining_set_1", "COPY-GROUP", "group_dining_set_0", {}],
        ["group_dining_set_1", "BACK-OF", "group_dining_set_0", {"
    distance": 0.4}],
        ["group_dining_set_1", "FACING-OPPOSITE-TO", "group_dining_set_0
    ", {}],
        ["group_dining_set_1", "ALIGN-CENTER-LR", "group_dining_set_0",
    {}]
```

```
                    ]
Other notes:
      The copied group will automatically have the same set of objects. For
       example, if group_dining_set_0 have ["plate_0", "fork_0", "knife_0",
       "cup_0"], then group_dining_set_1 will have ["plate_0-
      group_dining_set_1", "fork_0-group_dining_set_1", "knife_0-
      group_dining_set_1", "cup_0-group_dining_set_1"].
       No need for extra predicates for individual objects in the new group,
       they will keep their relation the same as the copied set.

2-7. SPECIAL:
PLACE-IN
Purpose: Place object A or a set/group of objects A inside container B.
      Useful for items in baskets, boxes, pen-holders, etc.
       Usage: [A, "PLACE-IN", B, {}]
       Conventions:
            B must be a container or support object that can hold items.
            A can a specification list of new objects by category and
      quantity, e.g. [["pen", 6], ["pencil", 3]] or [["pen", 3]]. In this
      case, the external function creates the specified items and places
      them.
            If you are placing multiple object into a container, please make
      sure you only use place-in once to put those objects into the
      container rather than calling separate times.

Examples:
     [[["pen", 6], ["pencil", 3]], "PLACE-IN", "pen_holder_0", {}]

PLACE-ANYWHERE
Purpose: Place object A anywhere in the scene without specifying
      relationships to other objects.
The position is determined by an external function that ensures physical
      plausibility (no collision, falling, etc.).
Example:
     ["vase_0", "PLACE-ANYWHERE", "root", {}]
Other notes:
      Useful for free-placement items or when exact position is not
      critical. And for placement in crowded scene.
      Do not use place-anywhere for a group.
      Because it does not depend on other objects, PLACE-ANYWHERE should
      appear at the end of a placement round, after all relational
      predicates.

3. NOTES FOR THE AGENT
* The support surface is already placed in the scene, with "root" as its
      name.
* For each object, please first give a one-sentence description for asset
       retrieval, in the format of a list of length 2: ["object_id", "
      descriptions"]
* An object is fully solved when one of {min_x|center_x|max_x} (FRONT-OF,
       BACK-OF, ALIGN-CENTER-FB, ALIGN-FRONT, ALIGN-BACK, SYMMETRY-ALONG,
      PLACE-ON or PLACE-ON-BASE with params can confirm this),
                                   one of {min_y|center_y|max_y} (LEFT-OF,
      RIGHT-OF, ALIGN-CENTER-LR, ALIGN-LEFT, ALIGN-RIGHT, SYMMETRY-ALONG,
      PLACE-ON or PLACE-ON-BASE with params can confirm this),
                               height (PLACE-ON-BASE, PLACE-ON),
                               and yaw (rotational direction) are
      determined (FACING-TO, FACING-SAME-AS, FACING-OPPOSITE-TO, SIDE-SCALE
      -ALIGN, RANDOM-ROT can confirm this).
                                   Special relationships (PLACE-IN, SELF-
      STACKING) can confirm everything through external functions.
```

```
* The predicate of the same object serving as A should be a segment in
    the list (group the predicate for that object together in the
    sequence).
* When you introduce a new object X_N, any relationship [X_N, predicate,
    Y, ...] must refer to Y that either is "root" or was introduced
    earlier in this list. Do not refer to objects that come later.
* Output must be a plain list of predicates, nothing else.
For example:
[
    ["laptop_0", "a slim silver laptop with a minimalist design"]
    ["laptop_0", "PLACE-ON-BASE", "root", {"x": 0.0, "y": 0.0}],
    ["laptop_0", "FACING-SAME-AS", "root", {}],

    ["notebook_0", "a medium-sized notebook with lined pages"]
    ["notebook_0", "PLACE-ON-BASE", "root", {}],
    ["notebook_0", "FRONT-OF", "laptop_0", {"distance": 0.1}],
    ["notebook_0", "RANDOM-ROT", "root", {}],

    ["cup_0", "an empty ceramic cup with a handle"]
    ["cup_0", "PLACE-ON", "notebook_0", {"overlap": 1.0}]
    ["cup_0", "FACING-FRONT", "root", {}]
    ...
]
* If two or more objects or sub-arrangements share the same relative
    pattern (same relative distances/orientations), the assistant should
    create a group for that pattern and use copy-group to replicate it,
    to ensure consistency and brevity.
* Naming convention is {category}_{identifier} like candle_0 or
    candle_front.
* Special predicates always comes alone to decide the placement of one
    object, do not couple with other predicates for one object.
* Pay attention to the sequence of predicates. It should be first other
    predicates, then PLACE-ON predicates and the predicates related to
    PLACE-ON objects, and finally PLACE-ANYWHERE predicates.
```

Listing 1: The system prompt for generating predicates.

```
{
    "role": "system",
    "content": f"{system_prompt}"
}
{
    "role": "user",
    "content": f"The xy extend of the table is {table_bbox}.\n The scene
    description is \"{scene_prompt}\"."
}
## if previous response and feedback exist ##
{
    "role": "assistant",
    "content": f"{previous_response}"
}
{
    "role": "user",
    "content": f"There are some errors in previous response. Here's the
    feedback {feedback}. Please generate a new one to fix it and try to
    retain the existing relationships if possible. You should still
    strickly follow the output format."
}
## end if ##
```

Listing 2: The complete prompting context for generating predicates.

### A.4.3 DETAILS ON PLACE-ON AND PLACE-ANYWHERE IMPLEMENTATION

We implement the solver for `place-on` and `place-anywhere` with an occupancy-grid-based heuristic augmented by a physics engine. We convert objects and the current scene into occupancy grids using `libigl` (Jacobson et al., 2018), with the grid resolution set to 0.03 for ground scene placement and 0.01 for other scenes. After voxelization, we use `correlate` from *scipy* (Virtanen et al., 2020) to find all non-penetrating positions between the object grid and the scene grid.

Then, we find the bottom surface of the object and the contact surface. The bottom surface is a 2D grid where each cell is considered part of the bottom if any one of the bottom $k_{bottom}$ voxels at that position is occupied in the object grid. The contact surface is defined as follows: from the bottom surface, each voxel searches below for $k_{search}$ voxels, and if a scene voxel is found, it is considered a contact. In most cases, we set $k_{bottom}$ and $k_{search}$ to 1 to ensure stability and organization. However, for a random scene, such as placing a box full of boxes and cans, we set both values to 5 to allow for more messy and natural placement. With the contact surface, we calculate a convex hull that surrounds all contact surface voxels, and the placement is considered valid if the projection of the center of mass of the object falls within the convex hull.

After all placement candidates are found, we rank them based on the parameters specified in the predicates, such as the closeness to the specified relative pose or the specified support ratio (which is defined as the area of the support surface over the area of the bottom surface), and choose the one that fits the best as the object's initial placement.

After all objects are assigned an initial placement, we run a physics simulation that allows all objects to free fall. Only objects with no large displacements are regarded as successfully placed, and their poses after the simulation will be their final poses.

### A.4.4 DETAILS ON STABILITY MEASUREMENT

We measure the stability of an object's placement by injecting perturbations into its physical parameters and validating the balance of the batch samples via simulation. Specifically, we model a configuration by a perturbation vector $x = \begin{bmatrix} \Delta p, \Delta r, \Delta c, \Delta \mu, \Delta m \end{bmatrix} \in \mathbb{R}^d$ with a nominal value $x_0 = \mathbf{0}$, where $\Delta p \in \mathbb{R}^3$ is a 3D position shift, $\Delta r \in \mathbb{R}^3$ is a small-angle rotation (axis–angle) shift, $\Delta c \in \mathbb{R}^3$ is a center-of-mass shift, $\Delta \mu \in \mathbb{R}$ is a friction-coefficient change, and $\Delta m \in \mathbb{R}$ is a mass change (so $d = 11$). Each dimension has a preset standard deviation $\theta \in \mathbb{R}_d$, giving a diagonal covariance $\Sigma = \operatorname{diag}(\theta_1^2, \ldots, \theta_d^2)$. We draw $N$ perturbations and label them via a physics engine: $x_j \sim \mathcal{N}(\mathbf{0}, \Sigma)$, $y_j = \mathbf{1}\{\text{fall under } x_j\} \in \{0, 1\}$, yielding $\mathcal{D} = \{(x_j, y_j)\}_{j=1}^N$. For any query $x$ (e.g., $x = \mathbf{0}$), we define the Mahalanobis distance $d_M(x, x_j)^2 = (x_j - x)^\top \Sigma^{-1}(x_j - x)$ and kernel weight $w_j(x) = \exp(-\frac{1}{2} d_M(x, x_j)^2)$; the weighted failure and total counts are $s(x) = \sum_{j=1}^N w_j(x)\, y_j$ and $n(x) = \sum_{j=1}^N w_j(x)$, giving the local failure probability $p_{\text{fail}}(x) = s(x)/n(x)$, which gives us a sense of the stability level. In addition, we can repeatedly optimize these parameters to find a highly unstable state by choosing the most-unstable yet still-stable configuration among the simulated points at each iteration, letting $S = \{i : y_i = 0\}$ and selecting $x^\star = \arg\max_{i \in S} p_{\text{fail}}(x_i)$; one may then re-center at $x^\star$ and repeat the process to refine.

### A.4.5 ADDITIONAL IMPLEMENTATION DETAILS

We use the `o4-mini` model as the LLM agent for generating predicates and asset descriptions. For the spatial solver, we use `Shapely` (GEOS contributors, 2024) to compute convex-hull overlaps in batch. The solver runs for 10 iterations; in each iteration, we loop through each parameter and uniformly sample 40 candidate values around the current estimate. Sampling ranges depend on the scene scale: we set distance parameters to one-tenth of the shortest scene dimension and rotation parameters to $\pm 10°$. Since convex-hull computations can be batched, the full process completes in under 5 seconds. For the physical solver, we use `Genesis` (Authors, 2024) as the physics engine.

### A.4.6 EVALUATION METRICS IMPLEMENTATION

We primarily adopt three quantitative evaluation metrics: VQA Score, GPT Ranking, and Settle Distance. Prior to evaluation, we re-render all scenes generated by different baselines to ensure consistent lighting conditions, viewing angles, and rendering parameters. For the **VQA Score**, we input

the rendered scene image into GPT-4o with the prompt "Does this image show {scene_prompt}? Please answer with Yes or No." and measure the probability of the "Yes" token in the response. For **GPT Ranking**, for each prompt, we aggregate the generated examples from all baselines and feed them to GPT-4o, requesting a ranking based on scene quality and alignment with the prompt. For **Settle Distance**, we initialize objects in a physics simulation with standard gravity according to their generated configurations. We run the simulation for 400 steps and calculate the displacement of each object between its initial and final positions. The displacement distance is clamped to a maximum of 1 (to account for objects falling off the table), and we report the average across all scenes.

## A.5 Experiment Details

### A.5.1 Baseline Implementation Details

For Architect (Wang et al., 2024b), we use their official implementation together with our own 3D asset database. 3D-Generalist (Sun et al., 2025b) is not open-sourced, so we re-implement their *Asset-Level Policy* following the instructions in the paper. The core of their pipeline is Molmo (Deitke et al., 2025), an open-source VLM that can directly output pixel coordinates. We use the same prompt as presented in their paper, and integrate Molmo with our grid-based search module to check whether the model's predicted placement is collision-free and supported. For LayoutVLM (Sun et al., 2025a), we also use their official implementation. Note that in their codebase, LayoutVLM requires a text prompt paired with a list of specified 3D assets as input. To adapt this method to our text-only scene generation setting, we introduce an additional pipeline that queries GPT-4o to generate a list of object descriptions, and then uses CLIP features to retrieve corresponding 3D assets from the database. To ensure a fair comparison, we process our 3D asset database (also sourced from Objaverse) into the format required by LayoutVLM.

### A.5.2 Analysis of Ablation Study on Feedback System

As shown in Table 2, our full feedback system significantly outperforms **Ours w/o report**, demonstrating the effectiveness of explicit spatial information feedback in guiding the correction process. Moreover, **Ours w/o report** outperforms **Ours w/o feedback**, which highlights the value of the other feedback components (i.e., Grammar Feedback and Solver Failure Feedback). These provide crucial information for the LLM-Agent to adjust its actions beyond what a simple binary signal can offer.

In addition, we tested visual feedback for failure recovery as an optional component. While the **Ours Visual** variant reduces the number of retries, it introduces additional time overhead due to API calls and rendering (note that rendering time is already excluded from the results in Table 2). Making this component optional allows our LLM-Agent to operate using only text input, which offers greater flexibility in model deployment.

### A.5.3 Analysis of Ablation Study on Placement Design

The results presented in Table 3 show that scenes generated by our framework have significantly better perceptual quality (VQA Scores and GPT Ranking) than both the **Random** and **LLM-Only** baselines. This demonstrates the importance of our predicates and solvers in creating semantically coherent arrangements.

In terms of physical plausibility (Settle Distance), the **Random** baseline performs well because its implementation ensures placement on a supporting surface and avoids complex stacking. The **LLM-Only** baseline, which lacks any explicit boundary or collision checks, performs the worst on this metric. These findings confirm that our method successfully ensures both high perceptual quality and physical stability in the generated scenes.

### A.5.4 Stacking Control Analysis

We demonstrate controlled scene generation with varying stability levels. As shown in Fig. 15, our method can generate both stable configurations (e.g., cups stacked securely inside one another) and fragile ones (e.g., a cup precariously balanced on top of another). This capability extends to

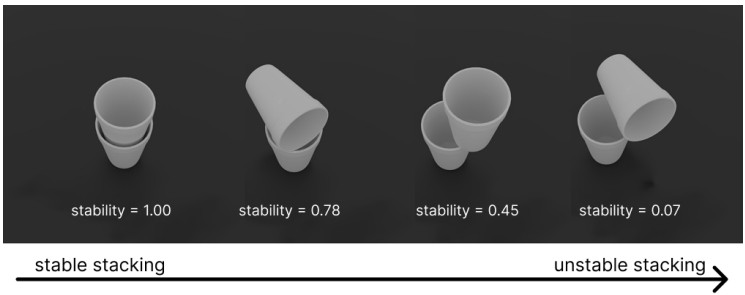

Figure 15: Generated placements for different stability level.

diverse objects and more complex arrangements such as cans, stacks of dishes, bowls, and utensils, as illustrated in Fig. 1 and Fig 3.

Our probabilistic programming framework, combined with Bayesian optimization, enables fine-grained control over object position, rotation, and physical properties to target more stable or unstable states. As shown in the stacking examples, we can adjust objects' centers of mass within a feasible range to create visually implausible yet physically realizable unstable states, where more than 90% of sampled perturbations lead to collapse. This ability to intentionally generate unstable scenarios provides a valuable tool for constructing challenging benchmarks and datasets for robotic manipulation.

### A.5.5 ROBOT DEMONSTRATION GENERATION

We build an automatic pipeline to generate robot manipulation demonstrations for our robot experiment. The general automated manipulation data-generation pipeline consists of four main components: (1) **Grasp sampler** — uniformly samples 5000 grasp candidates on the object and filters out invalid ones by lifting the object with a flying gripper in a physics simulation, serving as a pre-processing stage for each object. It then further filters valid grasps by checking their reachability with the robot arm; (2) **Goal state generator** — samples goal states based on robot-arm reachability and semantic constraints. Specifically, we consider two types of semantic relations: *on top of* and *inside*. Each relation has a predefined distribution over relative poses. For *on top of*, we sample goal poses on top of the target object's bounding box. For *inside*, we sample rotations such that the object's longest dimension aligns with the z-axis and allow overlapping between object bounding boxes. After sampling poses, we check validity by verifying that the robot can reach each pose with a valid grasp via inverse kinematics (the valid grasp provide relative pose between the end-effector and the object) and that the pose is penetration-free; (3) **Verifier** — after generating several valid goal states, we let the scene settle by running a physics simulation after the gripper releases the object, and then determine whether the result is a successful final state using a primitive verifier function combined with a VLM-based verifier. The primitive verifier exposes functions such as bounding box queries and collision checks, and we prompt an LLM to write a higher-level verifier function using these primitives. For the VLM verifier, we render the initial scene, the final scene, and a close-up view of the target objects in the final scene (with the targets highlighted with borders) and ask the VLM whether the task has succeeded. We label a state as successful only if both verifiers agree; (4) **Motion planner** — given the object's initial pose, the grasp pose, and the goal pose for release, we use cuRobo to plan trajectories both for reaching the grasp pose and for placing the object at the goal state after a successful grasp, thereby generating the final robot trajectory.

