# OpenReview forum: "PhyScensis: Physics-Augmented LLM Agents for Complex Physical Scene Arrangement"
_ICLR.cc/2026/Conference — ICLR 2026 Poster_

### Official Review · Reviewer_ummC · 2025-10-20

**Soundness:** 2
**Presentation:** 3
**Contribution:** 2
**Rating:** 4
**Confidence:** 4

**Summary:**

The paper presents a scene generation system for complex 3d scenes that prompts an LLM to produce scene descriptions using a scene predicate domain-specific language. Candidate predicates are processed by a 2d layout constraint solver and 3d physics engine to optimize and evaluate the scenes. Feedback from the DSL generation (syntax) and solvers is used to enable the LLM to iterate toward functional scenes with physical properties like asset placement instability.

Experiments test against two prior systems for 3d scene layout and ablations of some of the system components. Evaluations compare whether scenes match prompts, preference comparisons when evaluated by GPT, and the physical distances assets move after initialization (as a physical check). There is also an evaluation of model learning from demonstrations on these scenes.

**Strengths:**

# originality
The primary novelty is integrating multiple solver types as optimization and feedback mechanisms. This extends prior works and shows how to integrate more parts into generation and LLM guidance.

# quality
Shows some promising results on learning from scene demonstrations.


# clarity
Provides ample qualitative examples of the method to complement quantitative results.


# significance
Will be of interest to the robotic manipulation community.

**Weaknesses:**

# originality
No single component of the system is particularly novel. And the use of a DSL is more constrained than the more generic code generation of the 3DGeneralist prior work. None of this is horrible, but limits novelty.

# quality
See the questions for detailed remarks and suggestions. The primary concerns are:
- (1) Lack of statistical testing for differences and their magnitudes.
- (2) Need for scaling analysis to get a clearer sense of the cost-benefit trade-off of the new approach.
- (3) Lack of clarity on the demonstration generation and training process. The results are strong, but this is marred by ambiguity on how much the task reflects a particularly strong scenario for PhyScensis compared to previous efforts (the dinner table setting task).

# clarity
See the questions for minor comments. The demonstrations point (3) is related.

**Questions:**

# questions
- Table 1, 2, 3: Results should include statistical tests for differences and effect sizes. Some of the outcomes look to have overlapping standard deviations, suggesting the differences may not be large.
- Section 4.3: How were demonstrations generated: by humans? an automated process?
	- The section on demonstration generation and training is very compressed and hard to follow. I was not clear on what the demonstrations were, what training was done, and how evaluation was done.
- What costs are involved in each method evaluated (including ablations) and how do they scale?
	- For example: how many LLM queries used, how many iterations / computation (for solvers), how much wall clock time?
	- How do these costs scale with the scene size or number of assets? Other relevant input or output parameters?
- Table 2: Why are the VQA Score, GPT Ranking, Settle Distance not included?
- Is there any evidence around output scene diversity? How that impacts learning the outcomes?
	- It's often desirable that a generator can produce many different outputs from the same prompt, but this can be in tension with controllable outputs.
	- These metrics could be computed from the output scenes themselves, to measure things like number of assets, asset diversity across generations, placement diversity, and so on.


# suggestions / minor comments
- Figure 1: Why does "more compact" look the same?
- How would the dock scenario shown in the figures be used for manipulation?

---

> ### Author Response · Authors · 2025-11-21
> **Response to Reviewer ummC (Part 1)**
>
> *We appreciate your positive and insightful comments! Below, we address your concerns in detail.*
>
> **1. No single component of the system is particularly novel. And the use of a DSL is more constrained than the more generic code generation of the 3DGeneralist prior work. None of this is horrible, but limits novelty.**
>
> As discussed in General Response Section 1,
> our main contribution and novelty lie in how we incorporate a physics engine into the generation pipeline and how we design the feedback system, achieving physics accuracy and high efficiency.
>
> Regarding DSLs, 3D-Generalist also adopts one, as they state: “We use a domain-specific language (DSL) designed to represent 3D environments flexibly using a combination of code and natural language.” Our DSL is not more constrained: using PLACE-ON-BASE, one can directly specify any 2D location, and with FACING predicates, any facing direction. Thus, in principle, our DSL is as expressive as free-form code generation while still providing a structured interface for scene specification.
>
> **2. Lack of statistical testing for differences and their magnitudes.**
>
> We understand that overlapping standard deviations might imply marginal gains. We use GPT-4o for computing VQA scores, which tends to output values close to either 1.0 or 0.0, leaning to high variance. To address this, we performed paired t-test and calculated Cohen's d effect sizes on the test set to verify both the significance and the magnitude of our improvements, summarized as below:
> - Ours vs. 3D-Generalist: The difference is statistically significant ($p = 0.0055$, $t=2.94$) with a medium-to-large effect size ($d = 0.65$).
> - Ours vs. Architect: The difference is statistically significant ($p = 0.0001$, $t=4.34$) with a large effect size ($d = 0.94$).
>
> These metrics indicate that despite the variance inherent in the evaluation metric, our method provides a substantial and statistically robust improvement over the baselines
>
> **3. Need for scaling analysis to get a clearer sense of the cost-benefit trade-off of the new approach.**
>
> We present a run-time analysis in General Response 2.4. Statistically, our method is much faster than the two baselines when generating scenes with similar level of complexity.
> While generating physically complexed scene, like the one shown in Figure.1 (a)(3), indeed takes a long time with more iterations, it's worthy for a corner case generation.
>
> For scaling up scene generation (including for robot data collection in the paper), we first generate a scene using our full pipeline. Once a scene is successfully generated, we obtain its DSL description, including all object relationships, which we know to be valid. We then randomly jitter the numerical parameters in this DSL and re-run the spatial solver to obtain new scenes. This procedure produces similar, collision-free scenes with varied object positions, and each re-solving process takes on average only a few seconds.
>
> **4.Lack of clarity on the demonstration generation and training process. The results are strong, but this is marred by ambiguity on how much the task reflects a particularly strong scenario for PhyScensis compared to previous efforts (the dinner table setting task).**
>
> The manipulation task is designed to show that our generated scenes better align with human intuition compared to baseline methods, demonstrating both stronger text-following ability and higher overall scene quality.

---

> > ### Author Response · Authors · 2025-11-21
> > **Response to Reviewer ummC (Part 3)**
> >
> > > **Q4: Table 2: Why are the VQA Score, GPT Ranking, Settle Distance not included?**
> >
> > For the experiments in Table 2, our primary goal is to evaluate the time cost and the number of retries required to obtain a valid DSL under different design choices. These design variations should not significantly affect the final scene quality, since the rest of the pipeline remains identical and only resamples when a scene is invalid (i.e., violates constraints), rather than when it is merely less preferred.
> >
> > > **Q5: Is there any evidence around output scene diversity? How that impacts learning the outcomes?
> > It's often desirable that a generator can produce many different outputs from the same prompt, but this can be in tension with controllable outputs.
> > These metrics could be computed from the output scenes themselves, to measure things like number of assets, asset diversity across generations, placement diversity, and so on.**
> >
> > We argue that it is not essential for our method to explicitly optimize for diversity under a fixed prompt, since the approach is inherently open-vocabulary and can easily leverage diverse prompts generated by LLMs. For an ambiguous prompt like “a study desk,” we can simply ask an LLM to produce multiple, more specific variants with different details, which naturally lead to diverse scenes. In addition, diversity can be substantially increased by enlarging the asset library used for retrieval, making prompt- and asset-level variation a more meaningful lever than enforcing diversity at the method level.
> >
> > However, **there is evidence that our method can still produce diverse placements even under the same prompt and asset set**.
> >
> > 1. Some qualitative result of our method's ability to generate different scenes with the same prompt is shown in our website.
> > 2. As shown in Figure 12, our method can generate different placement but all resembles to the text description.
> > 3. To qualitatively compute the diversity of generated scenes under the same prompt, we first render and encode them into DINO features. Then we calculate the mean pairwise cosine distance and average 5-nearest-neighbor cosine distance between the embeddings. Specifically, mean pairwise cosine distance captures the global spread of the feature space by averaging the cosine distances across all pairs of scene embeddings. A higher value indicates that scenes differ more broadly from one another, implying greater overall diversity.
> > In contrast, the average 5-nearest-neighbor cosine distance measures local diversity by evaluating how similar each scene is to its closest neighbors. For each embedding, we find its five nearest neighbors and compute the mean cosine distance to them. Higher values suggest that even the most similar scenes still differ meaningfully, while lower values indicate clustering and redundancy within the generated set.
> >
> > |  | Ours | 3D-Generalist | Architect |
> > | -------- | -------- | -------- | --- |
> > | mean_pairwise | **0.2847** | 0.2579 | 0.1399 |
> > | avg_5nn | **0.2104** | 0.1651 | 0.0734 |
> >
> >
> > > **Figure 1: Why does "more compact" look the same?**
> >
> > The distance between the utensils and between the fork and the plate becomes closer, which makes the scene more compact.
> >
> > > **How would the dock scenario shown in the figures be used for manipulation?**
> >
> > We do not currently use it for any manipulation tasks; it is included purely as a demonstration that our method can extend beyond tabletop or other local scenarios. In the future, this capability could be leveraged for humanoid-robot manipulation.
> >
> > *We sincerely appreciate your comments. Please feel free to let us know if you have further questions.*
> >
> > Best,
> > Authors

---

> > > ### Comment · Reviewer_ummC · 2025-11-28
> > >
> > > Thank you for the responses! They help clarify many questions I had. These are the remaining questions to me:
> > >
> > > > These design variations should not significantly affect the final scene quality, since the rest of the pipeline remains identical and only resamples when a scene is invalid (i.e., violates constraints), rather than when it is merely less preferred.
> > >
> > > I think this should be shown with evidence, not assumed.
> > >
> > > > We do not currently use it for any manipulation tasks
> > >
> > > The paper should include what is used.
> > >
> > > > To qualitatively compute the diversity of generated scenes under the same prompt, we first render and encode them into DINO features
> > >
> > > This was a great way to address the diversity question!
> > >
> > > > The detailed of the general automated manipulation data generation pipeline consist of
> > >
> > > I found this explanation hard to follow as well. I think a cleaner version and appendix entry would help.
> > >
> > > > These metrics indicate that despite the variance inherent in the evaluation metric, our method provides a substantial and statistically robust improvement over the baselines
> > >
> > > Great!
> > >
> > > > Run Time Analysis
> > >
> > > This was welcome.

---

> > > > ### Author Response · Authors · 2025-12-02
> > > > **Further Response**
> > > >
> > > > Thanks the reviewer for acknowledging that many concerns are addressed!
> > > >
> > > > For the remaning questions:
> > > >
> > > > **1. Scores of Table 2:**
> > > >
> > > > |      | VQA-Score | GPT-Ranking | Settle Distance |
> > > > | ---- | --- | -------- | -------- |
> > > > | Ours w/o feedback | 0.657 ± 0.448| 2.34 ± 0.98 | 0.002 ± 0.006 |
> > > > | Ours w/o report | 0.682 ± 0.439| 2.74 ± 1.37 | 0.001 ± 0.005 |
> > > > | Ours Visual | 0.735 ± 0.414| 2.28 ± 1.52 | 0.003 ± 0.008 |
> > > > | Ours | 0.704 ± 0.425| 2.64 ± 1.03 | 0.003 ± 0.008 |
> > > >
> > > > We performed statistical comparisons of VQA-Score between our method and the three ablations. The results are as follows: Ours vs. Ours Visual ($t = -0.53, p = 0.59$), Ours vs. Ours w/o report ($t = 0.60, p = 0.54$), and Ours vs. Ours w/o feedback ($t = 0.24, p = 0.80$). None of these differences are statistically significant. The ranking differences are also relatively small. Since all configurations enforce physical correctness, the settle distance remains similarly low across all methods.
> > > >
> > > > **2. Scenarios used for manipulation:**
> > > >
> > > > Since the primary focus of this paper is scene generation rather than robot manipulation data generation, we restrict our manipulation experiments to the Robot Experiment section. As described there, we generate additional scenes for this experiment using the prompt “*a dining table set up for four people*” and execute a single manipulation task: “*pick up the leftmost cup and place it on the rightmost plate.*”
> > > >
> > > > We also randomly generate several tabletop scenarios and generate robot pick-and-place trajectories for them, as shown in the videos on our project website. However, all other generated scenes are not used for robot demonstration generation in this paper.
> > > >
> > > > **3. Automated Manipulation Data Generation Pipeline:**
> > > >
> > > > Briefly, we sample and verify candidate grasps in simulation, similarly sample and verify target states using a VLM and primitive checks, and finally use a motion planner to generate the execution trajectory. A more detailed description of this pipeline is provided in Appendix A.6.5.

---

> ### Author Response · Authors · 2025-11-21
> **Response to Reviewer ummC (Part 2)**
>
> ### Questions
>
> > **Q1: Table 1, 2, 3: Results should include statistical tests for differences and effect sizes. Some of the outcomes look to have overlapping standard deviations, suggesting the differences may not be large.**
>
> Since we use GPT-4o as the underlying model for computing VQA scores, it tends to output values very close to either 1.0 or 0.0 (i.e., highly deterministic judgments). As a result, the variance of these scores is large for all methods.
>
> > **Q2: Section 4.3: How were demonstrations generated: by humans? an automated process?
> The section on demonstration generation and training is very compressed and hard to follow. I was not clear on what the demonstrations were, what training was done, and how evaluation was done.**
>
> We make the manipualtion data generation part neat since it's irrelevant to our main content and contribution.
>
> The detailed of the general automated manipulation data generation pipeline consist of: (1) a grasp sampler, that sample and validate possible grasping pose for objects by a parallel simulation, (2) a goal state generator, which sample goal state based on robot-arm reachability and semantic (e.g., will sample a pose on plate and in the mean time guarentee reachable by the robot in some grasping pose), (3) a verifier, consist of a primitve verifier function and a VLM verifier, which will collaborate to confirm if the end state is a success one, (4) a motion planner, which we use Curobo to plan the trajectory of reaching to the grasping pose and placing the object to the goal state after the successfully grasped.
>
> It can be used to generate for any tasks, as the videos shown in the bottom of our website.
>
> As claimed before, we can generated a batch of scenes with a success DSL, and each batch of scenes is parallizable in terms of motion planning. We will use the same relative grasp pose and relative end state (cup relative to plate) with in a batch of scene. We collect in total 30 batches with 10 scenes in each batch and one trajectory for each scene for our methods. For baselines, the process is the same except that they can not be batched.
>
> For training, we use Lerobot diffusion policy, with a third person camera image and current robot state as input and the action would be the control target of joint positions of the robot for each step and the control frequency is set to 20 FPS.
>
> For evaluation, we collect 10 human placed table top scenes given the instruction, and run the diffusion policy to see if the generated trajectory succeed in this scene. We observe that the generated policy tend to perform the same with different samples in one scene, so we report the number of scenes the policy can succeed in. We further report reaching successful rate indicating that the network can recognize the target but fail to execute precisely the trajectory.
>
>
> > **Q3: What costs are involved in each method evaluated (including ablations) and how do they scale?
> For example: how many LLM queries used, how many iterations / computation (for solvers), how much wall clock time?
> How do these costs scale with the scene size or number of assets? Other relevant input or output parameters?**
>
> We present a runtime analysis in General Response 2.4. When generating scenes of similar complexity, our method is faster than the two baselines. In this experiment, 3D-Generalist requires one LLM call to generate the object list, followed by one VLM call for each object placement. In contrast, PhyScensis requires one primary LLM call, followed by 2–3 additional calls only if the solver fails. After each call, the batch of objects is resolved simultaneously. Consequently, when generating a scene with a large number of assets, PhyScensis scales better due to its batch solver. Architect is a pipeline that relies primarily on image inpainting models, depth prediction, and feature extraction. As a result, it is difficult to explicitly control the scene's complexity (e.g., the number of objects).

---

### Official Review · Reviewer_1kHh · 2025-10-29

**Soundness:** 2
**Presentation:** 3
**Contribution:** 2
**Rating:** 4
**Confidence:** 3

**Summary:**

The paper introduces PHYSCENSIS, a framework that automatically generates interactive and physically plausible 3d environments for robotic manipulation. PHYSCENSIS leverages an LLM to propose realistic scene configurations, including spatial relations and object properties. A physics solver then checks if the proposed configuration is feasible and places the objects in the scene. If objects are not solvable, a feedback will be provided to the LLM to refine the scene configuration. Experiments show that PHYSCENSIS outperforms two baselines and that the environments can be used to train an IL policy.

**Strengths:**

- The paper is very well written, well motivated, and easy to follow
- The methodology, although not entirely novel, is promising.
- The results and ablations show that the individual design choices result in improved generation speed and scene quality.

**Weaknesses:**

### The main weakness of the paper is the experiments. In particular, the downstream experiment fails to showcase the advantages of the approach compared to existing scene generation pipelines in the robotics domain:
- The VQA-based evaluation is questionable. It’s not clear if this metric works well for complex 3D tabletop environments. The high variance across models suggests it may not be reliable. Comparing it with human judgments could help validate this.
- It’s unclear whether the same VQA model and scores are used both for evaluation and for providing feedback during generation. If so, this would bias results in favor of PHYSCENSIS, since it would directly optimize for the evaluation metric.
- The chosen baselines are rather weak. The authors should explain why they did not compare against similar LLM + physics-based methods (e.g., ClutterGen, RoboGen, SimGen) and elaborate on the choice of baselines further
- The downstream manipulation task is too simple and does not demonstrate the framework’s claimed advantages. The task does not depend on accurate physics or object properties. More challenging tasks like stacking, unstacking, and manipulating objects with different stability would provide stronger evidence.
- The authors state that the cup and plate are fixed for each scene. Is this also the case for the baselines? During evaluation, are the plate and cup also fixed? If yes, the policy would not need to rely on visual cues.


The proposed method is not entirely novel, but it combines existing approaches. However, the problem is very relevant, and the framework could allow for training policies more robust to difficult settings in the pick and place task. However, in its current state, the experiments fail to showcase the effectiveness of the framework in that regard. Thus, in its current state, I tend towards reject. Performing extended robot experiments in more diverse environments and clarifying the evaluation methodology would strengthen the contribution and further support the paper's claims.

**Questions:**

See above and
- How does the framework compare against other frameworks in runtime?

---

> ### Author Response · Authors · 2025-11-21
> **Response to Reviewer 1kHh (Part 1)**
>
> *We appreciate the positive and insightful comments from you! We adress your concerns in details below.*
>
> **1. The VQA-based evaluation is questionable. It’s not clear if this metric works well for complex 3D tabletop environments. The high variance across models suggests it may not be reliable. Comparing it with human judgments could help validate this.**
>
> It has always been hard to evaluate generated scene quality in all prior works. Such criterias are used by prior works 3D-Generalist and Architect.
> We agree that human judgments would help and have added a user study in General Response Section 2.3.
>
> **2. It’s unclear whether the same VQA model and scores are used both for evaluation and for providing feedback during generation. If so, this would bias results in favor of PHYSCENSIS, since it would directly optimize for the evaluation metric.**
>
> We use the same VQA model and scoring procedure as in PHYSCENSIS generation (both GPT-4o). However, in our framework, the VQA feedback is primarily used to control the organized–messy and other property of the scene rather than its resemblance to the text prompt.
> Consequently, we disable this component of the feedback system in our quantitative experiments, which still makes it a fair comparison.
>
> **3. The chosen baselines are rather weak. The authors should explain why they did not compare against similar LLM + physics-based methods (e.g., ClutterGen, RoboGen, SimGen) and elaborate on the choice of baselines further.**
>
> To the best of our knowledge, the scene generation process in RoboGen [1] does not include a physics engine in the loop; it directly uses the LLM to output object poses. We further augment this with a simple collision-avoidance mechanism to form one of our LLM-only baselines.
>
> ClutterGen [2] does not enforce semantic coherence in its generated scenes and requires training for placing a fixed asset set, which is not directly comparable to the open-vocabulary, training-free nature of our method and baselines. Nevertheless, it is highly relevant in that it also targets physically accurate, cluttered scenarios. We therefore include an additional experiment comparing against ClutterGen; details are provided in General Response Section 2.2 and Appendix A.3.2.
>
> SimGen [3] focuses on driving-scene generation and is therefore less directly related to our setting.
>
> Overall, to the best of our knowledge, 3D-Generalist and Architect are the strongest recent baselines that explicitly address semantically coherent small-object scene generation with dedicated modules, which is why we treat them as our primary comparisons.
>
> **4. The downstream manipulation task is too simple and does not demonstrate the framework’s claimed advantages. The task does not depend on accurate physics or object properties. More challenging tasks like stacking, unstacking, and manipulating objects with different stability would provide stronger evidence.**
>
> The manipulation task is intended to demonstrate that our generated scenes more closely align with human intuition compared to baseline methods. We also show on our project webpage that, using our scenes, **we can generate action trajectories for complex tasks such as stacking cups or picking a pen from a cluttered pen holder**.
>
> While we aim to tackle more challenging downstream tasks in future work, getting a reliable robot policy in these scenarios is extremely difficult and would require substantially more generated data, which is beyond the scope of a scene-generation paper. Moreover, designing, training, and evaluating such advanced policies is not realistically achievable within the rebuttal period.

---

> ### Author Response · Authors · 2025-11-21
> **Response to Reviewer 1kHh (Part 2)**
>
> **5. The authors state that the cup and plate are fixed for each scene. Is this also the case for the baselines? During evaluation, are the plate and cup also fixed? If yes, the policy would not need to rely on visual cues.**
>
> By *“we fix the cup and plate assets across scenes”*, we mean that we use the same cup and plate mesh in all 300 scenes to simplify the task setting; this is also true for the baselines. Even though the assets for cups and plates are identical, their positions vary from scene to scene, so the policy must still rely on visual cues to localize the leftmost cup and rightmost plate. Refer to Figure 12 for more examples about generated test scene.
>
> **6. The proposed method is not entirely novel**
> As claimed in General Response Section 1, the main novelty of our work lies in the design of the feedback system and the incorporation of a physics engine into the generation loop, which together enable complex and physically accurate scene generation.
>
> **7. How does the framework compare against other frameworks in runtime?**
>
> We present a run time analysis in General Response 2.4. Experimentally, our method runs much faster than the baselines when generating scenes of similar complexity.
>
>
> [1] Wang, Yufei, et al. "Robogen: Towards unleashing infinite data for automated robot learning via generative simulation." arXiv preprint arXiv:2311.01455 (2023).
> [2] Jia, Yinsen, and Boyuan Chen. "Cluttergen: A cluttered scene generator for robot learning." 8th Annual Conference on Robot Learning. 2024.
> [3] Zhou, Yunsong, et al. "Simgen: Simulator-conditioned driving scene generation." Advances in Neural Information Processing Systems 37 (2024): 48838-48874.
>
>
> *We hope the additional explanations have convinced you of the merits of our work.*
> *We appreciate your time! Thank you so much!*
>
> Best,
> Authors

---

### Official Review · Reviewer_MLGL · 2025-10-31

**Soundness:** 4
**Presentation:** 4
**Contribution:** 3
**Rating:** 6
**Confidence:** 4

**Summary:**

The paper addresses the task of generating physically plausible environments. To tackle challenges in both spatial arrangement and physics, the authors propose the PhyScensis framework, which leverages a large language model (LMM) to generate predicates and employs a physics engine as the physics solver. Their framework also incorporates feedback from the physics engine back to the LMM for further refinement, resulting in realistic layouts and physically stable scenes. Experimental results demonstrate superior performance compared to previous methods.

**Strengths:**

* The overall system, which integrates LLM-based predicates, a physics-based solver, a geometry-based spatial solver, and feedback to the LLM, is well-designed. This results in layouts that are both reasonable and physically stable.
* Physics-plausible scene generation is an interesting and important direction, particularly for large-scale scene generation.
* The experiments are thorough, including ablations and additional evaluations on downstream robotics tasks.

**Weaknesses:**

* It is unclear what text prompts are used in the test set for all methods. How many prompts are there, and how diverse are they?
* There is no discussion of failure cases, particularly regarding physics. What are the limitations of the current predefined predicates?
* Regarding the LLM, it is unclear how it determines object sizes and how it selects objects from the candidate object set.

**Questions:**

* How are objects selected—only at the category level, or is there a more detailed retrieval?
* Are there predefined rules for selection, or is it random? For example, in the “table for 4” case, why are all plates the same? Is this constraint imposed by the LLM?

---

> ### Author Response · Authors · 2025-11-21
> **Response to Reviewer MLGL**
>
> *We appreciate the positive and constructive comments from you! We have modified our paper according to your comments.*
>
> **1. It is unclear what text prompts are used in the test set for all methods. How many prompts are there, and how diverse are they?**
>
> We prompt an LLM to generate a diverse set of 50 scene descriptions for testing, and for each description we generate 10 scenes for quantitative evaluation. Please refer to General Response Section 3.2 for sample scene descriptions.
>
> **2. There is no discussion of failure cases, particularly regarding physics. What are the limitations of the current predefined predicates?**
>
> We present a detailed failure analysis in Appendix A.2, including retrieval failure, solution not found, and unstable simulation, which reveals some of the limitations of our framework. The most frequent failure regarding physics is that the scene will become  unstable when many objects are added, and in this case a valid placement might not be found since objects will always fall.
>
> **3. Regarding the LLM, it is unclear how it determines object sizes and how it selects objects from the candidate object set.**
>
> For objects in the asset library, we assume their scales are known and fixed. For newly introduced objects, we prompt an LLM to predict the length of the largest dimension and rescale the object accordingly.
>
> For object retrieval, the LLM first generates a textual description for each object. We then encode these descriptions using OpenAI text embeddings and CLIP embeddings, compute their cosine similarity with the embeddings of objects in the asset library, and select the objects with the highest similarity scores.
>
> ### Questions
>
> > **Q1: How are objects selected—only at the category level, or is there a more detailed retrieval?**
>
> We use a more detailed retrieval procedure. Specifically, the LLM first generates a textual description of the desired object. We then compute OpenAI text embeddings and CLIP embeddings for this description and retrieve the closest matches from our dataset based on embedding similarity. If no object exceeds a similarity threshold, we instead perform 3D generation of the object directly from the description.
>
> > **Q2: Are there predefined rules for selection, or is it random? For example, in the “table for 4” case, why are all plates the same? Is this constraint imposed by the LLM?**
>
> There are cases with hard constraints and cases without.
>
> In the *“table for 4”* example, the LLM may produce predicates such as:
>
> ```
> [
>     ["plate_0", "a ceramic plate"],
>     ["plate_0", "PLACE-ON-BASE", "root", {"x": -0.2, "y": -0.2}],
>     ["plate_0", "FACING-SAME-AS", "root", {}],
>     ["cup_0", "an empty ceramic cup with a handle"],
>     ...
>     ["group_dining_set_0", "GROUP", ["plate_0", "fork_0", "knife_0", "cup_0"], {"anchor": "plate_0"}],
>     ["group_dining_set_1", "COPY-GROUP", "group_dining_set_0", {}],
>     ["group_dining_set_1", "FRONT-OF", "group_dining_set_0", {"distance": 0.4}],
>     ...
> ]
> ```
>
> Here, the scene is created by copying a group, so each copied place setting is *guaranteed* to use the same objects.
>
> In contrast, if the LLM uses separate object instances:
>
> ```
> [
>     ["plate_0", "a ceramic plate"],
>     ["plate_0", "PLACE-ON-BASE", "root", {"x": -0.2, "y": -0.2}],
>     ["plate_0", "FACING-SAME-AS", "root", {}],
>     ["cup_0", "an empty ceramic cup with a handle"],
>     ...
>     ["plate_1", "a ceramic plate"],
>     ["plate_1", "PLACE-ON-BASE", "root", {"x": 0.2, "y": -0.2}],
>     ["plate_1", "FACING-SAME-AS", "root", {}],
>     ["cup_1", "an empty ceramic cup with a handle"],
>     ...
> ]
> ```
>
> there is no hard constraint that `plate_0` and `plate_1` (or `cup_0` and `cup_1`) must be identical assets. However, by common sense, table settings are usually uniform, so the LLM tends to generate identical or very similar textual descriptions, which in turn leads the retrieval module to select the same underlying objects.
>
>
> *We hope the additional explanations have convinced you of the merits of our work. Please do not hesitate to contact us if you have other concerns.*
>
> *We really appreciate your time! Thank you!*
>
> Best,
> Authors

---

### Official Review · Reviewer_LKXm · 2025-11-01

**Soundness:** 2
**Presentation:** 3
**Contribution:** 2
**Rating:** 4
**Confidence:** 3

**Summary:**

The paper proposes a framework that leverages LLMs and a physics engine to generate physically plausible scenes. Specifically, given a library of 3D assets and a language caption, an LLM is first used to iteratively propose relevant assets and predicates that determine their initial positions. Then, spatial and physics solvers are used to ensure the layout is collision-free and physically plausible. Experiments show that the model achieves good performance compared with previous baselines, especially on scenes with cluttered objects.

**Strengths:**

1. The paper is well written and easy to follow.
2. The generation results look good.
3. The proposed method enables a certain level of controllability, such as the distance between objects and the stability of objects.

**Weaknesses:**

1. I think the term scene generation used here is misleading. The paper mostly focuses on “object arrangement” [1] or “layout generation” [2], where the goal is to place objects of similar sizes on a given surface (e.g., a bookshelf or a table). This is implied by all the qualitative examples. In contrast, scene generation usually refers to generating larger and more complex indoor scenes containing objects of various sizes and more diverse object relationships, which is not demonstrated in the experiments. I agree that the paper tackles a challenging problem involving arranging a large number of objects in confined space, but it is different from the indoor scene generation problem that the baselines address. If the authors want to keep using this term and still treat Architect / 3D-Generalist as the main baselines, they should provide more results on indoor scenes to demonstrate the effectiveness of the method.
2. LayoutVLM [2] is an important missing baseline. This model uses a similar asset library to specify spatial relations between objects. It would be useful to compare these methods. Unlike PhyScenesis, which only leverages an LLM and a physics simulator to avoid collisions, LayoutVLM prompts a VLM to get initial object positions, which may yield more semantically meaningful results.
3. The technical contribution is limited. SceneCraft [3] also proposes an agentic framework that generates 3D scenes using an LLM to generate Blender code with a feedback loop. Spatial and physical predicates are also used in LayoutVLM [2]. The design of the spatial and physics solvers here is largely heuristic and feels ad hoc. They seem specifically designed for cluttered scenes with small objects, which may limit applicability to other scenarios like full indoor scene generation or settings with fewer objects.
4. I appreciate the authors’ effort in building such a complex system to achieve good results. However, it would be better to provide more experiments and details to systematically justify the design choices. See my questions below for more information.

Minor Point:
Line 106: SceneThesis also has a module for optimizing the physical plausibility of generated scenes, including collision avoidance and stability.

References:
1. Line 106, Scenethesis also has a module for optimizing the physical plausibility of the generated scene including collision avoidance and stability.

[1] LEGO-Net: Learning Regular Rearrangements of Objects in Rooms. Qiuhong Anna Wei, et al. CVPR 2023 (Missing citation)

[2] LayoutVLM: Differentiable Optimization of 3D Layout via Vision-Language Models. Fan-Yun Sun, et al. CVPR 2025 (Missing citation)

[3] SceneCraft: An LLM Agent for Synthesizing 3D Scene as Blender Code. Ziniu Hu, et al. ICML 2024 (Missing citation)

I am happy to raise my score if my questions and concerns are addressed.

**Questions:**

1. Many experimental details are missing:

   a. How many prompts are used in the experiments, and how were they created?
   b. How many examples were generated per prompt?
   c. How exactly are the VQA score and GPT-based ranking implemented?

2. From the method description and the prompt in A.3.2, it seems the proposed method does not support placing objects on other objects at a specified height (e.g., a certain shelf level). How is this achieved in Figure 5?

3. The robot experiment is an interesting way to show the diversity of generated scenes from a single prompt:
   a. Why is the success rate of reaching much higher than that of placing?
   b. How are the human-designed test scenes different from the generated scenes?
   c. What do the generated scenes from each model look like? It would be helpful to show sample visualizations to illustrate quality and diversity.

4. What are the failure cases of the model? I am especially interested in scenes with low VQA scores or GPT rankings.
5. It would be helpful to provide qualitative examples of the ablation study between Random, LLM-Only, and the full method. What about using only the spatial solver without the physics solver?

---

> ### Author Response · Authors · 2025-11-21
> **Response to Reviewer LKXm (Part 1)**
>
> *Thank you for your insightful and constructive comments! We have added additional experiments and modified our paper according to your comments.*
>
> **1. Scene Generation Term**
> While our work focuses on arranging objects on a given surface, the objects are not necessarily of similar size, and the relationships between them can be considerably more complex than in prior works. Architect and 3D-Generalist explicitly separate large furniture from small objects, modeling only simple “supporting” relationships; their large-furniture arrangement is also driven largely by 2D spatial constraints. As a result, their modules does not yield a relationship graph more complex than ours, which handles a richer hierarchy of spatial relationships—including supporting, containing, and other fine-grained interactions.
>
> Our framework is also modular and can be integrated with any existing method for global furniture or room layout generation: prior systems can produce the coarse scene structure, and our method can then generate high-fidelity small-object placements. We compare to Architect and 3D-Generalist because both include dedicated modules for small-object placement on surfaces, separate from their furniture-layout pipelines. As discussed in General Response Section 2.2, we also compare to ClutterGen, which specifically targets tabletop clutter generation.
>
> **2. LayoutVLM Baseline**
> We show the LayoutVLM baseline results in General Response Section 2.1 and Appendix A.3.1. Thanks for mentioning that.
>
> **3. Technical Contribution**
> As highlighted in General Response Section 1, our core contribution and novelty lie in the efficient integration of a physics engine into the generation pipeline and the design of a more comprehensive feedback system, which together guarantee physical accuracy and improve efficiency.
>
> We do not claim to be the first to introduce an agent-based framework for scene generation, nor the first to consider physics plausibility. However, previous agent frameworks for scene generation [1, 2] rely solely on rendered images and VLM-based feedback, limiting the agent’s ability to accurately perceive and reason about physical interactions. Likewise, prior works aiming to improve physics plausibility [2, 3, 4] typically focus only on simplified constraints such as collision avoidance or supporting contacts, without incorporating full physical properties such as stability, friction, and other dynamics. These limitations restrict their ability to generate complex arrangements while maintaining true physical accuracy. Our method is designed to fill this gap.
>
> **4. SceneThesis also has a module for optimizing the physical plausibility of generated scenes, including collision avoidance and stability.**
> Its stability constraint only encourages an object to make contact with its supporting surface, without incorporating any additional physical constraints. As a result, the generated placements are not guaranteed to remain stable when evaluated in a full physics simulation.
>
> [1] Hu, Ziniu, et al. "Scenecraft: An llm agent for synthesizing 3d scenes as blender code." Forty-first International Conference on Machine Learning. 2024.
> [2] Sun, Fan-Yun, et al. "3D-Generalist: Self-Improving Vision-Language-Action Models for Crafting 3D Worlds." arXiv preprint arXiv:2507.06484 (2025).
> [3] Ling, Lu, et al. "Scenethesis: A language and vision agentic framework for 3d scene generation." arXiv preprint arXiv:2505.02836 (2025).
> [4] Sun, Fan-Yun, et al. "Layoutvlm: Differentiable optimization of 3d layout via vision-language models." Proceedings of the Computer Vision and Pattern Recognition Conference. 2025.

---

> ### Author Response · Authors · 2025-11-21
> **Response to Reviewer LKXm (Part 2)**
>
> ### Questions
>
> > **Q1: Many experimental details are missing:
> a. How many prompts are used in the experiments, and how were they created?
> b. How many examples were generated per prompt?
> c. How exactly are the VQA score and GPT-based ranking implemented?**
>
> Please refer to General Response 3.1 for metrics implementation, and General Response 3.2 for experiment details.
>
> > **Q2: From the method description and the prompt in A.3.2, it seems the proposed method does not support placing objects on other objects at a specified height (e.g., a certain shelf level). How is this achieved in Figure 5?**
>
> We first identify all supporting surfaces of the shelf (as there may be multiple layers) and then generate object placements for each layer. We adopt similar method for 3D-Generalist and LayoutVLM baseline.
>
> > **Q3: The robot experiment is an interesting way to show the diversity of generated scenes from a single prompt:
> a. Why is the success rate of reaching much higher than that of placing?
> b. How are the human-designed test scenes different from the generated scenes?
> c. What do the generated scenes from each model look like? It would be helpful to show sample visualizations to illustrate quality and diversity.**
>
> a) Reaching success indicates that the trained model correctly identifies the “leftmost cup.” In contrast, placing success requires highly precise grasping—millimeter-level errors can cause the grasp to fail, which is the dominant source of failure in our experiments. Notably, once an object is successfully grasped, all subsequent placing actions succeed.
>
> b) The human-designed scenes reflect natural human intuitions. Ideally, our generated scenes should be indistinguishable from these, as this would indicate human-level performance. To evaluate this, we train on our generated scenes and test on human-designed scenes, measuring the performance gap as an indicator of realism and generalization.
>
> c) We find that prior methods struggle to follow instructions due to limitations in both their image-inpainting components and their VLM-based pointing models, which often fail to precisely localize or manipulate the target objects. We've also shown 5 samples for each methods in Figure 12 in  paper Appendix.
>
> > **Q4: What are the failure cases of the model? I am especially interested in scenes with low VQA scores or GPT rankings.**
>
> We present failure results and detailed  analysis in Appendix A.2, including retrieval failure (Figure 7), solution not found (Figure 8), unstable stacking (Figure 9), and cases with low VQA Score (Figure 10).
>
> > **Q5: It would be helpful to provide qualitative examples of the ablation study between Random, LLM-Only, and the full method. What about using only the spatial solver without the physics solver?**
>
> We present qualitative examples of ablation study in Appendix A.2.1.
>
> Spatial solver and physical solver are for different predicates. Without physical solver, it would be not possible to generate scenes with any stacking or containing behaviors, limiting the complexity of the scenarios.
>
>
> *We wish that our response has addressed your concerns, and turns your assessment to the positive side. If you have any more questions, please feel free to let us know during the rebuttal window.*
>
> Best,
> Authors

---

### Author Response · Authors · 2025-11-21
**General Response to All Reviewers （Part 1)**

*We express our gratitude to all the reviewers for their perceptive comments aimed at enhancing the quality of our work.*

## 1. Our Contributions

Our primary contribution is introducing an efficient approach for integrating a physics engine into the scene layout generation agent, together with a comprehensive feedback mechanism that enhances the agent’s ability to perceive and reason about its environment.
This design address a core limitation of prior scene-generation methods—their inability to produce complex arrangements that are physically accurate.
By enabling reliable generation of scenarios such as tabletop arrangements, shelf organization, and box-packing layouts with assured physical fidelity, our method substantially advances the capabilities of existing systems.

## 2. Additional Experiments

**2.1. LayoutVLM Baseline**
LayoutVLM[1] adopts a similar method for 3D scene-level layout generation, using LLMs to generate proposals for object positions and relationships, followed by a differentiable solver to determine the final layouts. We present additional quantitative results in the following table. In Appendix A.3.1, we further present the implementation details, qualitative results, and detailed analysis for this baseline.


| Method | VQA Score ↑ | GPT Ranking ↓ | Settle Distance ↓ |
| :--- | :---: | :---: | :---: |
| Architect | 0.493 ± 0.392 | 3.250 ± 0.947 | 0.405 ± 0.471 |
| 3D-Generalist | 0.578 ± 0.399 | 2.393 ± 0.899 | 0.033 ± 0.048 |
| LayoutVLM | 0.648 ± 0.446 | 2.643 ± 1.109 | 0.115 ± 0.238 |
| Ours | **0.704 ± 0.425** | **1.714 ± 0.920** | **0.003 ± 0.008** |

*Table: Additional quantitative comparison of PhyScensis with baselines, including LayoutVLM.*


**2.2. ClutterGen Baseline**
ClutterGen[2] addresses a related problem of generating complex object arrangements on a supporting surface of a specified size. Their approach employs reinforcement learning to optimize a policy for placing a fixed set of assets, but it does not consider semantic coherence.
Thus, we compare the maximum number of objects that can be placed in a scene using the same asset library. Results show that ClutterGen places an average of only 3.57 objects per scene, whereas our method places an average of 14.77 objects without stacking and 22.30 objects with stacking. For additional details and qualitative examples, please refer to Appendix Section A.3.2 and Figure 14.


**2.3. User Study**
We conducted a user study comparing our method against two main baselines: Architect and 3D-Generalist. Specifically, we selected 6 prompts and randomly sampled one result for each of the three methods, creating a total of 18 evaluation cases. For each case, we asked users to rate the generated scene on a scale of 1-5 across three dimensions: text alignment, naturalness & physical plausibility, and complexity. We collected responses from 20 participants; the results are as follows:


| Baseline | Match with Text ↑ | Naturalness & Physics Plausibility ↑ | Complexity ↑ |
| :--- | :--- | :--- | :--- |
| **Architect** | 2.68 | 2.65 | 2.69 |
| **3D-Generalist** | 2.54 | 2.72 | 3.04 |
| **ours** | 4.04 | 3.98 | 3.82 |

Our method significantly outperforms the other two baselines, which is consistent with our quantitative metrics. This result also aligns with our qualitative observations: Architect estimates object layout directly from inpainted images, often leading to object penetrations and implausible physics. Meanwhile, 3D-Generalist employs a VLM to output placement positions in pixel space. However, the VLM's limited spatial reasoning capability fails to yield reasonable layouts for complex prompts, resulting in this baseline receiving the lowest scores for text alignment.


**2.4. Run Time Analysis**

We calculated the average runtime for generating 50 scenes. To ensure fairness, we configured 3D-Generalist and PhyScensis to generate a similar number of objects. Architect cannot control the specific number of generated objects because it utilizes an inpainting-based pipeline. Additionally, we excluded rendering time for baselines, as ray-tracing renderer is computationally intensive and independent of the generation process. The results are as follows:

| Baseline | Run Time | Number of Objects |
| :--- | :--- | :--- |
| **Architect** | 201.32 ± 2.47| 5.22 |
| **3D-Generalist** | 177.20 ± 21.57 | 12.15 |
| **ours** | 95.95 ± 30.23 | 11.82 |

Our method is the fastest of the three. Architect requires multiple rounds of inpainting, diffusion based depth prediction, feature extraction, and solving, resulting in a complex and heavy pipeline. 3D-Generalist requires a VLM call and a rendering step for each object placement, which limits its scalability. In contrast, our method resolves a batch of objects after a single LLM call, making it the most efficient. However, because our solver may occasionally fail—requiring the LLM to iteratively propose new predicates—there is a larger standard deviation in runtime.

---

### Author Response · Authors · 2025-11-21
**General Response to All Reviewers (Part 2)**

## 3. Additional Implementation Details

**3.1 Quantitative Metrics Implementation Details**

We primarily adopt three quantitative evaluation metrics: VQA Score, GPT Ranking, and Settle Distance. Prior to evaluation, we re-render all scenes generated by different baselines to ensure consistent lighting conditions, viewing angles, and rendering parameters. For the **VQA Score**, we input the rendered scene image into GPT-4o with the prompt ``Does this image show {scene_prompt}? Please answer with Yes or No.`` and measure the probability of the ``Yes`` token in the response. For **GPT Ranking**, for each prompt, we aggregate the generated examples from all baselines and feed them to GPT-4o, requesting a ranking based on scene quality and alignment with the prompt. For **Settle Distance**, we initialize objects in a physics simulation with standard gravity according to their generated configurations. We run the simulation for 400 steps and calculate the displacement of each object between its initial and final positions. The displacement distance is clamped to a maximum of 1 (to account for objects falling off the table), and we report the average across all scenes.


**3.2 Test Set Prompt Generation**

We prompt LLM to generate a diverse set of 50 scene descriptions for testing.

Here's a sample of 10 of them:
```
"rustic_still_life": "A wooden table with fruits, a candle, and old books arranged in a painterly still-life composition.",
"sunflower_artist_desk": "An artist’s desk with a vase of sunflowers, brushes, paints, and a sketchpad.",
"romantic_dinner": "A candlelit dinner setup for two with plates, glasses, and a warm atmosphere.",
"messy_breakfast": "A cluttered breakfast table with food, spilled drinks, and a phone left on the side.",
"fantasy_board_game": "A game table mid-session with cards, dice, miniatures, and snacks around.",
"model_train_workbench": "A hobbyist’s desk with train parts, tools, paints, and in-progress assembly.",
"traveler_memento_table": "A table with maps, postcards, foreign coins, and a globe—suggesting travel memories.",
"explorer_desk": "An explorer’s writing table with a compass, sketches, and an old map spread out.",
"science_experiment": "A chemistry experiment scene with beakers, colored liquids, and handwritten notes.",
"space_project": "A tabletop with star maps, a model rocket, small tools, and technical components."
```
Then we generate 10 scenes for each prompt for quantitative evaluation.


[1] LayoutVLM: Differentiable Optimization of 3D Layout via Vision-Language Models. Fan-Yun Sun, et al. CVPR 2025
[2] Jia, Yinsen, and Boyuan Chen. "Cluttergen: A cluttered scene generator for robot learning." 8th Annual Conference on Robot Learning. 2024.

---

### Meta-Review · Area_Chair_rA7X · 2026-01-06

**Summary:**

This paper introduces PhyScensis, which can generate physically plausible 3D scene layouts. It is an agent-based approach integrated with large language models. The authors highlight that the proposed approach accounts for physical relationships, such as contact, stability, and containment, which were overlooked in prior work.

**Reviewer Concerns:**

The authors raised the following concerns
1. Baselines (LKXm, 1kHh): The reviewer asks a comparison with LayoutVLM, ClutterGen
2. Evaluation metrics (1kHh, ummC): The main evaluation metric (VQA score) may not be the correct metric for 3D environment
3. Runtime and scaling (ummC): The reviewers ask about the cost-benefit trade-off
4. Statistical significance (ummC): The reviewer requested statistical tests.
5. Scope (LKXm): The reviewer mentioned that the proposed approach is rather "object arrangement" rather than "scene generation"
6. Simple downstream tasks (1kHh): The reviewer mentioned that the robotic manipulation task was too simple to justify the proposed approach.

**Reviewer Scores:**

The paper received mixed scores:

- MLGL: marginally above the acceptance threshold (with minor questions, such as prompts)
- ummC: marginally below the acceptance threshold
- LKXm: marginally below the acceptance threshold (mentioned that the score can be upgraded if the issue is resolved)
- 1kHh: marginally below the acceptance threshold

The authors provided an extensive rebuttal to resolve the concerns raised above. For instance, authors provide statistical tests and runtime analysis. The authors added comparisons with LayoutVLM and ClutterGen, showing better results. The authors conducted a user study to address concerns about the appropriate evaluation metric.

AC notes that the author's rebuttal and revision can resolve most of the major concerns raised by reviewers ummC, LKXm, and 1kHh. Therefore, AC recommends accepting the paper (with a possible bumped-down confidence). AC strongly agrees with reviewer LKXm's comment that the proposed approach is more of a 3D object arrangement than full 3D generation. The concerns by 1kHh about the simple downstream task also need to be more justified. Therefore, AC recommends revising the paper and title, stating that this is an 'arrangement task' rather than a 'generation task' and suggests more interesting downstream tasks.

---

### Decision · Program_Chairs · 2026-01-26

Accept (Poster)